# Physical Processing-Assisted pH Shifting for Food Protein Modification: A Comprehensive Review

**DOI:** 10.3390/foods14132360

**Published:** 2025-07-03

**Authors:** Ruiqi Long, Yuanyuan Huang, Mokhtar Dabbour, Benjamin Kumah Mintah, Jiayin Pan, Minquan Wu, Shengqi Zhang, Zhou Qin, Ronghai He, Haile Ma

**Affiliations:** 1School of Food and Biological Engineering, Jiangsu University, 301 Xuefu Road, Zhenjiang 212013, China; 2212318004@stmail.ujs.edu.cn (R.L.); 2112318160@stmail.ujs.edu.cn (Y.H.); pjiayin@163.com (J.P.); 19710512320@163.com (M.W.); zhangshengqi0418@163.com (S.Z.); 19599960979@163.com (Z.Q.); mhl@ujs.edu.cn (H.M.); 2Institute of Food Physical Processing, Jiangsu University, 301 Xuefu Road, Zhenjiang 212013, China; 3Department of Agricultural and Biosystems Engineering, Faculty of Agriculture, Benha University, Moshtohor, Qaluobia P.O. Box 13736, Egypt; mokhtar.dabbour@fagr.bu.edu.eg; 4CSIR—Food Research Institute, Accra P.O. Box M20, Ghana; b.minta20@gmail.com; 5Department of Agro-Processing Technology and Food Bio-Sciences, CSIR College of Science and Technology (CCST), Accra P. O. Box M32, Ghana

**Keywords:** pH shifting, ultrasonication, solubility, conformational attributes, plant protein

## Abstract

The increasing demand for sustainable protein sources has intensified interest in improving the processing efficiency of traditional proteins and developing novel alternatives, particularly those derived from plants and algae. Among various processing technologies, pH shifting has attracted attention due to its simplicity, low cost, and capacity to effectively alter protein structure and functionality. However, employing pH shifting alone requires extremely acidic or alkaline conditions, which can lead to protein denaturation and the generation of undesirable by-products. To address these limitations, this review explores the integration of pH shifting with physical processing techniques such as ultrasound, high-pressure processing, pulsed electric fields, and thermal treatments. Moreover, this review highlights the effects of these combined treatments on protein conformational transitions and the resulting improvements in functional properties such as solubility, emulsification, foaming capacity, and thermal stability. Importantly, they reduce reliance on extreme chemical conditions, providing greater sustainability in industrial applications, particularly in food product development where milder processing conditions help preserve nutritional quality and functional properties. In that sense, this combined treatment approach provides a promising and eco-efficient protein modification strategy, and bridges technological innovation with sustainable resource utilization.

## 1. Introduction

With the rapid growth of global population and shifts in dietary preferences, the demand for high-quality protein has significantly increased, placing pressure on existing protein supply chains [1]. Traditional protein sources are currently in supply–demand imbalance, and as a consequence the selection of protein for various industrial applications is undergoing a significant process of diversification. Animal-derived proteins, including whey, casein, and meat proteins, have extensively been used in providing nutrition due to their comprehensive amino acid profiles [2,3]. However, they are limited by low resource conversion efficiency (15–40%) and a high carbon footprint (accounting for 14.5% of total emissions). Moreover, plant-based proteins (e.g., soy, pea, and rice) are receiving much attention due to their bioavailability and sustainable production, yet challenges remain in terms of inconsistent extraction yields (40–68%) and difficulties in functional property modification [4,5]. Meanwhile, emerging biomass proteins such as algal proteins (e.g., spirulina, chlorella) and insect proteins (e.g., black soldier fly, beetle larvae) offer high reproduction efficiency (yield per unit area is up to 150 times greater than soybeans) and unique nutritional profiles. Despite these advantages, large-scale application and consumer acceptance continue to pose significant research challenges [4,6,7,8].

Recent studies have shown that the individual or combined use of physical, chemical, and biological modification methods can effectively improve the functional properties of proteins from various biological sources. These methods alter protein structures in targeted ways and enhance key functions such as solubility, emulsifying ability, and gel formation, which are crucial for both food and industrial applications [7,9,10,11,12]. Among various physical, chemical, and biological approaches for protein modification, the pH shifting method has recently emerged as a research theme of interest. Compared with traditional methods (including heat treatment, fermentation, etc.), it offers advantages such as simplicity, stability, and high efficiency. Moreover, its non-thermal mechanism, controllability, and environmentally friendly nature distinguish it from other methods.

The pH shifting technique has received considerable attention as a useful strategy in protein extraction and modification. By adjusting the acidity or alkalinity of solution, the pH shifting method effectively modifies protein structure, resulting in the augmentation of functional properties, thereby enhancing its application in industry [10,13,14]. In addition, the pH shifting method can enhance emulsifying properties, gelling ability, and thermal stability of proteins in a way that can increase the texture and sensory characteristics of food products [15,16]. Nevertheless, the use of the pH shifting technique presents certain practical limitations. Extreme pH conditions may cause protein denaturation, impair biological activity, and potentially lead to the formation of harmful by-products [12,17]. Also, the use of pH shifting alone for protein treatment often fails to yield the desired outcomes [18]. Therefore, to overcome this limitation, the combination of pH shifting techniques with physical modification methods, such as ultrasound, ultra-high pressure, and pulsed electric fields, has emerged as a critical research direction in recent years. The synergistic interactions resulting from these integrated methods can notably improve protein extraction efficiency and functional properties. Moreover, recent advancements in novel physical extraction techniques, including ultrasound-assisted extraction, ultra-high pressure processing, and pulsed electric field treatment, have significantly enhanced both extraction efficiency and protein purity [19]. Earlier studies show that protein structure can be modified using physical methods, thereby enhancing functional properties such as solubility and emulsification ability, and reducing allergenicity [20,21,22,23]. Compared to individual treatments, the combination of pH shifting with physical modification techniques often results in significantly enhanced functional properties of proteins, suggesting the presence of synergistic interactions. Notably, the use of physical methods in combination avoids the introduction of exogenous chemical agents. Moreover, the mechanisms involved in such combined treatments are relatively straightforward, making them more accessible for mechanistic interpretation [13,14,17].

Moreover, Momen et al. [24] presented a comprehensive review on the application of alkaline-mediated treatments for both the extraction and functional modification of proteins from plant and animal sources, and reported that alkaline conditions irreversibly unfolded protein structure, thereby improving solubility, emulsification, gelation, and bioactive compound binding properties. Additionally, Lou et al. [25] discussed the molten globule state of proteins induced by pH shifting. Their work focused on the structural characteristics of this intermediate state and its relevance to food processing applications, emphasizing the potential of partially unfolded proteins in improving emulsification, flavor retention, and gelation. Recently, Sultan et al. [26] provided an extensive review on the pH shifting technique for the extraction of plant-based proteins from various sources such as legumes, cereals, and oilseeds, and noticed that such a method had the ability to produce protein isolates with improved solubility, emulsification, foaming, and gelation properties. Concurrently, they systematically summarized the effects of extraction and precipitation pH on both protein yield and functionality, highlighting optimal pH ranges for various protein types. Furthermore, this review briefly mentioned the effect of pH shift combined with ultrasound on the structure and functionality of proteins, but did not provide a detailed explanation.

Although these studies have laid a foundation for understanding protein behavior under alkaline or pH-induced conditions, several limitations persist. including negligible functional improvement and extreme operational conditions. Notably, previous reports have largely overlooked the synergistic effects of combined pH shifting and physical processing techniques, such as high-intensity ultrasound (HIU), pulsed electric field (PEF), and high-pressure treatment. These emerging hybrid strategies may enable more precise control over protein unfolding, aggregation, or reconfiguration, thereby augmenting their functional characteristics. While pH shifting significantly enhances functional properties, such as solubility and emulsifying activity via structural changes (e.g., *β*-sheet dissociation, hydrophobic group exposure), its potential limitations should not be overlooked. Extreme pH conditions may induce chemical modifications of amino acid residues (e.g., serine, lysine), stimulating reduced nutritional quality and the formation of toxic compounds like lysinoalanine (LAL). While alkaline extraction can temporarily enhance the solubility of rice protein, overexposure to high pH levels (pH > 12) may result in irreversible deamidation and the loss of essential amino acids [5]. Therefore, precise control of the pH range and the development of multiscale synergistic modification techniques have become critical research directions.

Therefore, this review aims to provide an updated and comprehensive overview of the synergistic application of pH shifting and physical modification techniques in the structural and functional transformation of food proteins, from both plant and animal sources. The review systematically elucidates the physicochemical mechanisms governing protein conformational changes under extreme pH conditions, and elucidates their effects on secondary, tertiary, and quaternary structures. The review further explores how the integration of pH shifting and other physical techniques such as ultrasonication, microwave treatment, and pulsed electric field enhance key functional properties, including solubility, emulsification, gelling, foaming capacity, and bioactivity. Additionally, it discusses current and potential applications of these combined strategies in food processing. While critically addressing existing technological constraints, research gaps and future directions of combined pH-physical strategies for protein valorization in the food and agricultural industries were discussed.

## 2. Effect of pH Shifts on Proteins

The pH shifting treatment is a simple but effective chemical modification method. It operates by exposing proteins to highly acidic (pH < 2.5) or highly alkaline (pH > 10.5) conditions, which increases the surface charge of amino acids on protein molecules. This leads to enhanced electrostatic repulsion between protein molecules and promotes the unfolding of protein chains. Under ionic and dipole interactions, the binding between charged amino acids and water molecules increases, thereby improving protein solubility [27]. The treatment typically involves two stages. First, the protein solution is adjusted to an extreme pH (acidic or alkaline) and left to stand, allowing the elevated surface charge to induce molecular unfolding. Then, the pH is adjusted back to neutral (pH 7.0) and the solution is allowed to stand again. At this stage, the net surface charge decreases, weakening protein–water interactions, and hydrophobic forces drive the refolding and aggregation of protein molecules, often resulting in precipitation. During refolding, misfolding of protein chains can occur, altering the molecular structure and leading to functional modifications. The greater the degree of misfolding, the more pronounced the modification effect. This process induces various chemical reactions within protein molecules, ultimately altering their structure and enhancing functional properties. The pH shifting method is valued for its simplicity, precision, low cost, and significant modification efficiency. Only by adjusting the acidity or alkalinity of solution can one significantly improve protein solubility and functional properties such as emulsification, foaming, and thermal stability, while largely preserving the original structure and bioactivity.

Changes in pH significantly affect the structural stability and functional properties of proteins by modulating non-covalent interactions such as hydrogen bonds, ionic bonds, and hydrophobic interactions. At the molecular level, pH fluctuations induce the protonation or deprotonation of amino acid residues (e.g., His, Asp, Glu), directly altering hydrogen bond networks and ionic interactions. Under highly acidic conditions (e.g., pH < 3), *α*-helix structures become destabilized, leading to an increased content of *β*-sheet and random coil. Simultaneously, electrostatic repulsion is reduced due to charge shielding effects. Contrarily, alkaline pH may disrupt the stability of the hydrophobic core, resulting in the exposure of buried residues (Figure 1). This may cause partial unfolding of the tertiary structure, creating intermediate states with exposed thiol and hydrophobic groups, which provide a structural basis for subsequent aggregation or functional modifications. Figure 1 illustrates the changes in primary, secondary, and tertiary protein structures, as well as inter-molecular interactions, in response to the pH shifting process [28]. Under normal conditions, the primary structure of proteins remains stable after a pH shift, and the order of amino acids remains basically unchanged. However, under extreme pH shift conditions, the secondary structure usually shows that both the α-helix and β-fold undergo structural deconvolution and destruction of hydrogen bonding. There is also structural loosening and rearrangement, as well as an increase in random coils. Significant changes occur to the tertiary structure of proteins under pH shift conditions. The tertiary structure of proteins undergoes significant changes under pH shift treatment, mainly manifested by the destruction of salt bridges within the protein molecule and the exposure of hydrophobic groups and sulfhydryl (-SH) groups, which can form disulfide bonds (-S-S-) [29].

Given the critical role of pH shifting in modulating protein structure and function, relevant literature is summarized in Figure 2 to clearly illustrate the effects exerted by varying pH conditions on protein conformation. The phase behavior of ovalbumin gradually evolves with the decrease in pH, leading to formation of amorphous aggregates instead of gel bead-like aggregates, and spherulites instead of needle-like crystals [30]. For example, ovalbumin tends to form spherical aggregates rather than crystals at pH 4.7. However, moderate deviations from the isoelectric point (e.g., pH 3 or pH 8) enhance surface charge density, thereby improving protein solubility and dispersion. This property is of significant value in bio-separation processes. For instance, treating protein complexes under acidic conditions (pH 3–4) can increase solubility by 1.5–2 times compared to neutral pH, significantly enhancing extraction efficiency [31]. Moreover, plant storage proteins undergo quaternary structural destruction under extreme pH conditions (pH 1.5/12), generating soluble monomers and aggregates. The proteins after conformational changes exhibit excellent emulsifying and foaming properties due to side chain rearrangement. pH-induced structural transitions have successfully been utilized in food texture modulation and biomaterials engineering [32]. Notably, it was found that under extreme alkaline conditions (pH 11 and 11.5), protein solubility and recovery rates of rainbow trout by-products were significantly enhanced. Structural alterations included reduced particle size, decreased α-helix content, increased *β*-sheet and other secondary structures, and improved emulsification and foaming properties. These effects were further amplified under combined pH shifting and ultrasonication treatments [33]. Variation in pH conditions effectively alters protein structures, enhancing their functional properties such as solubility and emulsification, and exhibits synergistic potential when integrated with physical techniques like ultrasonication.

Recent studies have demonstrated that combining pH shifting with physical processing technologies (such as ultrasound, high-pressure, and pulsed electric fields) can overcome the limitations of standalone treatment. Ultrasound-induced cavitation can suppress excessive aggregation under extreme pH while enhancing interfacial adsorption and solubility (by over 50%). High-pressure treatment, by applying mechanical forces, cooperates with pH-induced conformational rearrangement to reduce disulfide bonds (with >90% retention of free thiol groups), thereby enhancing functionality and reducing LAL formation (by 60%). These synergistic strategies not only mitigate the side effects of the pH shifting process but also enable directed structural reconstruction through energy barrier modulation, offering innovative solutions for food (e.g., texture optimization of plant-based meat), pharmaceuticals (e.g., targeted delivery systems), and nutrition (e.g., development of high-bioavailability proteins). Therefore, a deeper exploration of such hybrid strategies holds great promise for advancing the precision and applicability of protein engineering in food systems [33].

## 3. Combination of pH Shifting and Physical Processing Techniques

### 3.1. Heat Treatment-Assisted pH Shifting

Temperature plays a crucial role in protein extraction and modification, with thermal treatment being one of the most commonly employed physical methods. By increasing temperature, thermal treatment enhances the interaction between protein and solvent molecules and increases the thermal kinetic energy of proteins. This disrupts non-covalent bonds within proteins including hydrogen bonds, hydrophobic interactions, and van der Waals forces to induce structural unfolding and rearrangement. Such controlled, heat-induced structural changes can significantly improve the functional properties of proteins, including solubility, emulsifying stability, and gelation ability.

When proteins are subjected to acidic or alkaline conditions far from their isoelectric point, the effects of heat-induced structural changes are further intensified. Under these conditions, significant alterations in surface charge distribution increase electrostatic repulsion, resulting in a looser protein structure. The thermal energy generated also enhances protein molecular mobility, accelerating the disruption of intramolecular non-covalent bonds and leading to further unfolding and exposure of functional groups. Consequently, many studies have combined controlled thermal treatment with pH shifting techniques to synergistically improve protein functionality and broaden their applicability in the food and agricultural sectors.

#### 3.1.1. Effect of Heat Treatment-Assisted pH Shifting on Protein Structure

Many reports have indicated that an extreme pH shift combined with heating induces substantial protein unfolding and restructuring of secondary/tertiary structures [34]. Under thermal-alkaline conditions, globular proteins tend to lose their compact tertiary structure as strong electrostatic repulsion and thermal energy cause the molecules to partially unfold. A common outcome is the cleavage of stabilizing disulfide bonds, evidenced by a significant increase in free sulfhydryl groups and dissociation of high-molecular-weight subunits [12,35]. These unfolded proteins undergo notable secondary-structure transitions including a considerable decrease in *α*-helix content accompanied by an increase in *β*-sheet (or *β*-turn) structures after pH shifting and heat treatment, indicating that some helical regions refold into *β*-sheet conformations or other element structures [36]. This transition coupled with the exposure of hydrophobic amino acid residues and reactive sites reflects a more flexible, partially denatured conformation of protein. Notably, the newly exposed functional groups (e.g., thiols and hydrophobic groups) can intensify the intermolecular interactions. Sun et al. [37] reported that alkaline heating induced unfolding of the tertiary structure of soy protein, resulting in the exposure of previously buried reactive sites. This structural change enhanced enzymatic cross-linking mediated by transglutaminase, ultimately leading to the formation of stronger protein gels. Such findings indicate that pH shifting combined with thermal treatment considerably altered protein structure, resulting in the collapse of disulfide linkages. These changes may have converted *α*-helix into *β*-sheet structure, forming a less compact conformation with more reactive protein state conducive to aggregation and improved functional properties [12]. pH shifting induces conformational changes by altering the surface charge of proteins, while thermal treatment provides kinetic energy that disrupts non-covalent interactions, facilitating molecular unfolding and stimulating the exposure of functional groups [38]. Under alkaline conditions combined with heat, globular proteins lose their compact tertiary structures due to strong electrostatic repulsion and thermal energy, leading to partial unfolding of protein [39]. This process frequently involves the cleavage of stabilizing disulfide bonds, resulting in increased free sulfhydryl groups and dissociation of high-molecular-weight protein subunits [40]. Additionally, proteins undergo secondary-structure transitions, specifically a decrease in *α*-helix content and an increase in *β*-sheet structures, suggesting a refolding from helical to extended conformation. Such structural changes expose hydrophobic amino acid residues and reactive sites, significantly enhancing protein functionality [41].

#### 3.1.2. Effect of Heat Treatment-Assisted pH Shifting on Protein Functionality

The combination of pH shifting and thermal treatment has recently emerged as a promising synergistic strategy for enhancing the functional properties of both plant and animal proteins. The enhanced extraction efficiency of protein is primarily attributed to improved solubility, which is directly caused by protein unfolding induced through pH shifting and heating. The micromorphology of Silkworm Pupa Protein Isolates (SPPI) was reported by Xu et al. [12] to be altered by alkaline pH shifting (pH 12.5) and heat treatment (80 °C) for 60 min, with the disulfide bonds between macromolecular subunits (72 and 95 kDa) being destroyed, resulting in reduced particle size and increased zeta potential and free sulfhydryl content of the isolates. The fluorescence spectra analysis showed red shifts with increased pH and fluorescence intensity improved with temperature (40–90 °C), implying the alterations in the tertiary structure of protein. This improvement may reduce the aggregation and increase the molecular flexibility following the cleavage of disulfide bonds and decrease in *α*-helix content, thereby augmenting protein solubility.

pH shifting combined with heat treatment can also improve the emulsifying properties of protein. Sun et al. [42] demonstrated that heat-assisted pH shifting treatment at pH 12 and 70 °C for 2 h significantly enhanced the emulsifying stability of pumpkin seed protein isolate (PSPI), achieving an internal oil phase volume fraction of up to 80%, and maintaining emulsion stability even after centrifugation at 10,000 g for 60 min and storage for 30 days. The data from sodium dodecyl sulfate-polyacrylamide gel electrophoresis (SDS-PAGE), UV–visible, intrinsic fluorescence, and Fourier transform infrared spectroscopy (FTIR) indicated that the primary, secondary, and tertiary structures of PSPI were disrupted, resulting in changes in interfacial tension, wettability, particle size, and dispersibility of protein. This may have promoted the adsorption of modified PSPI nanoparticles to the surface of oil droplets and improved the emulsion stability. In addition to emulsification, the combination of pH shifting and heat effectively enhances foaming properties, linked closely to protein unfolding and structural flexibility [42]. Chang et al. [36] demonstrated that combining pH shifting (pH 12, maintained for 1 h at room temperature) with controlled heating (70 °C for 30 min) significantly enhanced the foaming capacity of pea protein isolates. This improvement was mainly attributed to the formation of soluble protein aggregates, structural transformation from *β*-sheet to *α*-helix, and increased surface hydrophobicity, resulting in a more flexible protein conformation at the air–water interface. Protein unfolding increases flexibility and availability of hydrophobic sites that facilitate rapid adsorption at air–water interfaces, thereby stabilizing foam. The enhancement of gel properties is also driven by structural changes. Functional groups such as thiol groups are exposed through structural unfolding, promoting stronger intermolecular interactions and covalent cross-linking, thereby strengthening the protein network within the gel. Wang et al. [43] reported the changes in the structure and gel properties of peanut protein isolate (PPI1) under the synergistic effect of temperature and pH shifting. The breaking force and water-holding capacity of pH 10-treated PPI1 10–40 (at 40 °C) gel were 2.2 times and 2.15 times higher than that of the pH 7-adjusted sample at 25 °C. Moreover, the solubility, free sulfhydryl content, and surface hydrophobicity of PPI1 10–40 were 1.3 times, 1.8 times, and 1.6 times that of the pH 7-adjusted sample, respectively, which resulted in enhanced covalent or non-covalent interactions between proteins.

The combination of pH shifting and thermal treatment can significantly enhance functional properties of protein, including emulsifying ability, thermal stability, foaming capacity, and gelation performance. The existing literature suggests that this synergistic method not only improves protein functionality but also offers broad application potential in food, pharmaceutical, and related fields. Yang et al. [44] employed a combined pH shifting (pH 11) and thermal treatment (70 °C for 20 min) strategy to improve the functionality of whey protein isolate (WPI), successfully developing a novel method for preparing whey protein isolate-tryptophan (WPI-Trp) nanoparticles. Similarly, Sun et al. [42] successfully developed a food-grade Pickering emulsifier by employing a heat-assisted pH shifting treatment, specifically adjusting the pH to 12 and heating at 70 °C for 2 h followed by neutralization to pH 7, significantly enhancing the emulsifying properties of pumpkin seed protein isolate (PSPI) nanoparticles. Also, Zhong et al. [45] simulated gastrointestinal digestion and noticed that Pickering emulsions prepared using LP-Res nanoparticles under pH 11 and 60 °C (pH 11, 60 °C-LP-Res) effectively protected resveratrol (Res) and vitamin D3 from degradation and precipitation. Furthermore, Nisov et al. [46] reported that increasing the pH of raw plant protein materials, such as pea protein concentrate (PPC), pea protein isolate (PPI2), rice protein isolate (RP), and isolated wheat gluten (WG), to pH 7 prior to freeze-drying and extrusion processing at temperatures between 115 and 160 °C effectively enhanced the structural alignment and mechanical strength of the extrudates, thereby promoting their potential use as attractive plant-based meat analogues. Similarly, Zhu et al. [47] modified protein and starch mixtures by adjusting the pH to 8.0 and applying thermal treatment at 90 °C for 3 h. This treatment improved the emulsifying properties of rice starch and whey protein isolate conjugates, likely as a result of Maillard-type interactions and enhanced interfacial functionality.

The combined application of pH shifting and thermal treatment has been shown to significantly enhance protein functionality, including solubility, emulsifying capacity, foaming ability, and gelation. These improvements are primarily attributed to heat-induced protein unfolding and the exposure of reactive groups under extreme pH conditions, which promote intermolecular interactions and structural rearrangements. This strategy holds considerable potential for use in food formulation, functional delivery systems, and the development of plant-based protein products. Despite these advantages, several challenges persist. Most existing studies are conducted under controlled laboratory conditions, limiting their applicability to real food systems. Additionally, the outcomes are highly protein-specific and sensitive to processing parameters such as pH, temperature, and holding time. Excessive thermal treatment may also lead to undesirable protein aggregation, loss of nutritional value, and off-flavor formation. Therefore, optimizing processing conditions and elucidating the underlying molecular mechanisms are essential for broader industrial implementation. To address these limitations, researchers are investigating non-thermal approaches such as ultrasound-assisted pH shifting, which enables effective protein modification while minimizing thermal degradation.

### 3.2. Ultrasound-Assisted pH Shifting

Ultrasound treatment, a green and non-thermal physical processing technology, has widely been used in the extraction and modification of proteins [48]. Ultrasound, as a mechanical wave typically operating at frequencies above 20 kHz, has gained considerable attention across the food, medical, and industrial sectors due to its strong penetrability, absence of chemical contamination, and high controllability. It significantly improves protein extraction efficiency by disrupting the cellular matrix. Beyond facilitating extraction, ultrasound primarily alters the physical, structural, and functional properties of proteins through shear forces generated by cavitation. These forces disrupt non-covalent interactions such as hydrogen bonds and hydrophobic interactions, promoting protein unfolding or partial denaturation, and subsequently altering secondary and tertiary structures of protein [49]. As a result, protein solubility and reactivity are enhanced.

Moreover, ultrasound influences the surface properties of proteins by exposing hydrophobic regions, thereby enhancing their functionalities [50]. It can also disintegrate protein aggregates or precipitates, restoring proteins to their native states or improving their dispersibility, which further contributes to improved solubility and functional performance in food and other applications. However, ultrasound may also induce protein oxidation, aggregation, and cross-linking, which can negatively affect hydrophobicity, solubility, emulsification, and foaming properties [51]. Furthermore, ultrasound can alter various physical properties such as particle size, rheological behavior, conductivity, and ζ-potential, directly influencing the functional and nutritional quality of proteins [52].

The growing global demand for plant-based proteins has driven the need for efficient extraction techniques, making it a key research area in food science. Among these, ultrasound-assisted extraction (UAE) has gained significant attention due to its high efficiency, energy savings, and environmentally friendly characteristics. Figure 3 shows the effects of ultrasound and ultrasound assistance on the protein extraction rate and related structures. Studies have shown that UAE promotes the destruction of cell walls through cavitation, high shear force and mechanical energy transfer, allowing solvents to penetrate cells more easily, thereby enhancing the solubility of proteins and improving extraction efficiency [53]. For instance, in ultrasound-assisted alkaline extraction of pea protein isolate (PPI2), optimized conditions (solid-to-liquid ratio of 1:11.5 g/mL, pH 9.6, extraction time of 13.5 min, ultrasound amplitude of 33.7%) achieved a maximum PPI2 yield of 82.6%, significantly outperforming traditional alkaline extraction [11]. Additionally, UAE induced secondary and tertiary structural changes, causing partial unfolding and exposing hydrophobic groups, thereby improving solubility, emulsification, foaming capacity, and gel formation ability. For instance, in pecan protein extraction, an optimized ultrasound-assisted enzymatic method (400 W, 20 kHz, 5 s/3 s) improved protein yield to 25.51% by increasing substrate solubility and enzyme–protein interactions, accelerating chemical reactions and boosting protein recovery. This method also altered the secondary and tertiary structures of the protein, exposing hydrophobic groups and sulfhydryl (-SH) residues, which significantly enhanced solubility (70.77%), emulsification activity (120.56 m^2^/g), and dispersibility (0.305) [54]. In beer spent grain (BSG) protein extraction, UAE further demonstrated its potential for protein recovery and functional improvement. Under optimized conditions (250 W, 20 min, duty cycle 60%), BSG protein yield increased from 45.71% (traditional method) to 86.16% (UAE method). Structural analysis revealed that ultrasound treatment increased *β*-sheet content by 5.83% while reducing *α*-helix, *β*-turn, and random coil contents by 4.76%, 0.33%, and 0.74%, respectively, indicating conformational modifications [55]. In summary, UAE exhibits tremendous potential in plant protein extraction and functional modification, significantly improving yield, structural properties, and bioactivity. Future research should further investigate the underlying mechanisms of UAE and explore its synergy with other auxiliary techniques, such as high-pressure homogenization, electric field treatments, and enzymatic hydrolysis, to achieve more efficient protein extraction and modification.

Lately, the combined use of ultrasound and pH shifting has increasingly attracted interest in the field of protein modification. The synergistic mechanism of ultrasound-assisted pH shifting primarily relies on the complementary effects of both techniques: ultrasound-induced cavitation intensifies the structural alterations and reactivity of protein molecules, while pH shifting alters surface charge distribution, inducing further unfolding or rearrangement of protein structures. When proteins are exposed to acidic or alkaline environments far from their isoelectric point, surface charge and electrostatic repulsion increase, promoting chain relaxation and molecular unfolding. These structural changes are further amplified by ultrasound treatment, where the high shear forces generated by cavitation effectively disrupt non-covalent interactions within the protein, increasing structural unfolding and exposing more functional sites. Accordingly, the combined ultrasound and pH shifting treatment can significantly enhance protein functionalities, thus expanding their potential application in the food industry.

#### 3.2.1. Effect of Ultrasonication-Assisted pH Shifting on Protein Structure

The combined effects of ultrasound (US) and pH shifting on protein structure have (in recent years) emerged as an important research direction, particularly in relation to particle size, secondary and tertiary structures, surface properties, and intermolecular interactions. The structural alterations induced by such treatments depend significantly on the pH shifting that occurs under acidic or alkaline conditions. Ultrasound-assisted pH shifting has shown notable synergistic effects in protein modification, with the underlying mechanisms summarized as follows.

Firstly, electrostatic repulsion induced by pH shifting promotes protein unfolding, leading to the exposure of internal hydrophobic groups and reactive sites. Simultaneously, cavitation-generated microjets from ultrasound enhance chain extension and improve solvent penetration [56]. Secondly, under extreme pH conditions, hydroxyl radicals (·OH) produced by ultrasound selectively oxidize sulfur-containing amino acid residues, enabling controlled disulfide bonds reformation and covalent cross-linking [57]. Thirdly, when the pH is readjusted to near the isoelectric point, ultrasound suppresses irreversible aggregation and, through mechanical shear, regulates the refolding process, resulting in functional particles with uniform size and optimized surface properties.

Previous study displayed that pH shifting and sonication significantly increased the emulsifying activity index (by 62.8%) and thermal stability of soy protein isolate (*p* < 0.05) [58]. Research further indicates that ultrasound-assisted pH shifting disrupts both non-covalent interactions (e.g., hydrogen bonds and hydrophobic interactions) and covalent bonds (e.g., disulfide linkages), resulting in a more relaxed and unfolded protein conformation. Changes in secondary structure are particularly evident, with a general reduction in *α*-helix and *β*-sheet content, accompanied by increases in *β*-turn and random coil. Zheng et al. [59] demonstrated that the combination of ultrasound treatment at an intensity of 400 W for 15 min with pH shifting (adjusted to pH 12 and subsequently neutralized) at 25 °C significantly altered the secondary structure of soy protein isolate. Specifically, the contents of *α*-helix and *β*-sheet were markedly reduced, while *β*-turn and random coil structures were increased, indicating a transition toward a more disordered and flexible conformation. Similarly, Dabbour et al. [60] found that ultrasound combined with pH shifting significantly decreased *β*-sheet content and increased random coil and *β*-turn content in cottonseed meal protein, implying unfolding in secondary structure. These structural changes are often associated with increased exposure of sulfhydryl groups, cleavage of disulfide bonds, enhanced surface hydrophobicity and reactivity, and reduced particle size [61].

#### 3.2.2. Effect of Ultrasonication-Assisted pH Shifting on Functional Properties of Protein

Traditional physical and chemical methods have some limitations such as low efficiency, poor selectivity, and functional degradation. The application of ultrasound and pH shifting technologies has displayed significant advantages in enhancing extractability and functionality of protein. pH shifting and UAE have been indicated to effectively improve the extraction efficiency of both plant-based and marine by-product proteins, including pea protein isolate (PPI2), shrimp by-product protein, fish processing by-product protein, and oil body wastewater protein [62,63,64]. The cavitation effect and high shear forces generated by ultrasound facilitate the disruption of cell walls and the unfolding of protein structures, leading to an increase in solubility, thereby enhancing extraction efficiency [65]. For example, Yang et al. [66] reported that the ultrasound-assisted pH shifting treatment, in which the pH of pea protein isolate was adjusted to 12 followed by ultrasound application at 400 W and 20 kHz for 10 min at 25 °C, significantly improved protein solubility to over 90 percent. This was substantially higher than the solubility observed in samples treated with ultrasound alone or in the untreated control.

Moreover, the partial unfolding of proteins and the exposure of sulfhydryl and hydrophobic groups not only improve solubility but also enhance emulsifying activity, foaming capacity, gel-forming ability, and bioactivity [58,67]. For instance, proteins extracted from brewery spent grain using optimized UAE conditions exhibited significantly improved emulsifying, foaming, and fat-binding properties, attributable to conformational modifications in the secondary structure [8]. Table 1 shows the effect of pH shifting and ultrasound on functional properties of protein.

Ultrasound-assisted pH shifting protein modification holds great potential applications in functional ingredient delivery systems, enhancement of plant protein functionalities, and the design of novel food products. In the construction of delivery systems, this technique significantly improves the encapsulation efficiency of bioactive compounds by promoting the exposure of hydrophobic regions and the unfolding of protein structure. In a recent study, Liu et al. [81] demonstrated that combining pH 11 and 300 W ultrasound treatment for 20 min significantly enhanced the co-encapsulation of VE and QU in SLP matrices. This approach improved encapsulation efficiency, solubility, and antioxidant activity compared to untreated SLP, highlighting the potential of pH and ultrasound for modulating protein delivery systems. Similarly, Fang et al. [82] demonstrated that a combination of 540 W ultrasound-assisted pH shifting treatment at pH 12 for 5 min, followed by adjustment to pH 7.0, significantly enhanced the encapsulation efficiency of resveratrol in soy protein isolate (SPI) to 91.4 ± 4.3%. This result was substantially higher than the encapsulation efficiency achieved with ultrasound treatment alone (83 ± 3%) and significantly improved the functionality of SPI as a nanocarrier for hydrophobic compounds. In another study, Zhang et al. [71] fabricated solid high internal phase emulsions (HIPEs) with excellent elasticity and stability by modifying PPI2 with ultrasound (500 W, 10 min) at pH 12, followed by adjusting the pH to 7 to obtain modified PPI2 (MPPI2). MPPI2 and chitosan particles were used as emulsifier and co-stabilizer, respectively, to construct HIPEs with an enhanced interfacial adsorption and network structure, resulting in improved stability. These findings confirm the significant potential of ultrasound-pH synergistic modification in improving protein carrier performance, enhancing functional properties, and enabling the design of novel food formulations.

In summary, the combination of pH shifting and ultrasound alters protein conformation, enhancing solubility, emulsifying properties, and encapsulation capacity, thereby expanding their potential applications in bioactive delivery systems and emulsion-based food formulations. This aligns with previous discussions on the functional improvements of proteins induced by ultrasound-assisted pH modification.

### 3.3. High Pressure-Assisted pH Shifting

High-pressure (HP: 100–600 MPa) and ultra-high-pressure (UHP: >600 MPa) processing are non-thermal technologies that apply extreme mechanical forces to materials via hydrostatic pressure. HP treatment induces conformational alterations in protein by disrupting intermolecular non-covalent interactions such as hydrogen bonds and hydrophobic interactions [83]. Contrarily, UHP, due to its higher energy density, can additionally cause the disruption and reformation of covalent bonds, including disulfide bridges. Both techniques enable controlled structural modification of proteins at ambient or low temperatures while avoiding degradation of thermosensitive compounds. Consequently, they are widely applied in food, pharmaceutical, and other sectors for protein functionality enhancement and bioactivity preservation.

The combination of high-pressure and pH shifting is an effective strategy in protein modification and has extensively been used to improve functional properties of protein. Previous studies have exhibited that this combined approach effectively modulated protein conformation and enhanced its potential for delivering bioactive compounds [84,85,86]. High-pressure homogenization (HPH) can break disulfide-linked subunits into lower molecular weight fragments and convert highly ordered secondary structure into random coil, thereby facilitating the unfolding of protein [87]. In this regard, UHP has widely been adopted in the food processing industry, particularly for modifying biological macromolecules such as protein and starch. Under such conditions, UHP not only alters the higher-order structure of protein via physical interactions such as reducing the number of hydrogen bonds stabilizing protein structure, but also modulates their physicochemical properties, including solubility, particle size, and hydrophobicity [88]. These modifications ultimately enhance the functional performance of proteins in the food system by altering their structure [89].

#### 3.3.1. Effect of High Pressure-Assisted pH Shifting on Protein Structure

Presently, the synergistic application of pH shifting and high-pressure treatment, particularly HPH, has attracted significant attention in the field of protein modification. This combined approach modulates protein structure at multiple levels by adjusting the pH environment and applying mechanical pressure [90,91]. These structural modifications provide a molecular basis for enhancing the functional properties of protein. For example, the synergistic effect of HPH and pH shifting results in significant alterations in the secondary and tertiary structure of protein, inducing the release of free hydrogen ions, and thus increasing the surface positive charge density of protein [84]. The combination of pH shifting and HPH promoted the depolymerization of protein aggregates, reduced particle size, and decreased surface hydrophobicity [85]. Additionally, HPH-assisted pH shifting enhanced the structural flexibility of hemp protein isolate, as evidenced by the increased content of random coil. Literature shows that, following HPH and pH shifting, the intrinsic fluorescence intensity significantly decreased and exhibited a red shift, while surface hydrophobicity increased, implying extensive unfolding in protein conformation [92]. When the pH approached the isoelectric point, the *α*-helix structure was converted into *β*-sheet and *β*-turn by up to 20%; additionally, HPH alone also promoted the transformation of *α*-helix into *β*-sheet, with a conversion rate of 15% observed under extreme acid and alkaline conditions [93]. Moreover, Laguna et al. reported that pea protein at pH 6.2 demonstrated a higher degree and rate of digestion, whereas high-pressure treatment masked this pH-dependent effect, which was consistent with the observed structural changes [94].

Moreover, under optimal high-pressure conditions (600 MPa), the emulsifying activity of protein isolate was significantly enhanced, primarily due to protein unfolding and the increased exposure of hydrophobic groups [95]. In this context, pH variation serves a regulatory role in the high-pressure modification of proteins [92]. In the case of soy protein isolate, this treatment has been noticed to increase the content of free sulfhydryl groups and surface hydrophobicity, reduce droplet size, and enhance zeta potential [96]. For hemp protein isolate, high-pressure treatment resulted in an increase in random coil content and a decrease accompanied by a red-shift in intrinsic fluorescence intensity, indicating a significant enhancement in structural flexibility [92]. In addition, the HPH-assisted pH-shifting treatment enhanced protein penetration and rearrangement at the oil–water interface, which in turn facilitated the formation of a more stable interfacial layer [92]. pH shifting and high-pressure processing (HPP), particularly high-pressure homogenization (HPH), holds great potential in protein modification; this combined approach induces multi-scale structural transformations by altering the pH environment and applying mechanical stress. As a result, significant changes in protein secondary and tertiary structures have been observed, including increased random coil content, *α*-helix to *β*-sheet transitions, and enhanced structural flexibility. These conformational modifications lead to structure changes such as reduced particle size, altered surface charge, and increased hydrophobic group exposure, which collectively provide the molecular foundation for improved protein functionalities.

#### 3.3.2. Effect of High Pressure-Assisted pH Shifting on Functional Properties of Protein

This synergistic approach (e.g., pH shifting and high-pressure treatment, particularly HPH) has demonstrated significant advantages in enhancing solubility, emulsifying capacity, foaming properties, and interfacial activity. Zhu et al. [47] reported that a combination of pH shifting and high-pressure homogenization (HPH) treatment at a pressure of 500 MPa for five cycles at pH 3 increased protein solubility by 34.75%, and enhanced its binding affinity with vitamin B12, primarily through hydrogen bonding and hydrophobic interactions. Wang et al. [85] found that treatment of rice dreg protein with pH 12 and 20–100 MPa HPH increased solubility from 0.86% to 63.5%. Similarly, Yildiz and Yıldız et al. [97] observed significant improvements in solubility (from 7.85% to 78.97%) and soluble protein content in quinoa protein isolate by applying a sequential pH shifting treatment at alkaline pH 12 for 1 h, followed by high-pressure homogenization (250 MPa, single-pass) at ambient temperature. This combined approach effectively reduced particle sizes to 54 nm while enhancing surface hydrophobicity (198.0 ± 0.6) and antioxidant activity (9.8 ± 0.03%). Wang et al. [92] further observed that the solubility of hemp protein isolate reached a maximum of 62.8% following combined treatment with sequential pH shifting at alkaline pH 12.0 for 1 h, followed by high-pressure homogenization at 120 MPa for two consecutive passes at 25.0 ± 1.0 °C. This synergistic approach induced structural unfolding evidenced by a 2.2-fold increase in random coil content (from 44.2% to 98.3%) and particle size reduction from 1612 nm to 231 nm (*p* < 0.05), with treatment efficacy showing pressure-dependent optimization (R^2^ = 0.972). Wang et al. [85] achieved a 126.7-fold increase in emulsifying activity and a 930.7-fold increase in foaming capacity of rice dreg protein through sequential pH shifting (pH 12.0, 25 °C, 4 h) combined with three cycles of high-pressure homogenization under 100 MPa at 25 °C. Tan et al. [96] demonstrated that the synergistic effect of pH shifting (pH 11.0, 25 °C, 30 min) and high hydrostatic pressure (400 MPa, 10 min) significantly enhanced the emulsion stability of soy protein isolate (ESI from 21.8 to 35.1 min), accompanied by 51.5% reduction in droplet size and 132% increase in emulsifying activity index (EAI from 12.7 to 29.6 m^2^/g). Likewise, Ahmed et al. [95] also reported that under optimal conditions (600 MPa), high-pressure treatment alone markedly enhanced protein emulsification, with pH shifting playing a regulatory role in this process. Additionally, the synergistic application of pH shifting and high-pressure treatment substantially influenced the crystallization behavior and crystal morphology of protein, providing a molecular foundation for structural control and texture design in protein-based products [98]. Such modification has indicated broad applicability and promising potential across various plant proteins, including those derived from peas, rice dregs, soybeans, hemp seeds, and quinoa [53,99].

The application of pH shifting combined with high-pressure treatment in protein modification is primarily focused on areas such as bioactive compound delivery, novel food development, hypoallergenic food production, and overall improvement of food quality. For example, increased pH and high-pressure conditions significantly influence protein separation performance, particularly by affecting membrane permeability and salt rejection, thereby enabling the effective delivery of bioactive substances [100]. Khan et al. [101] found that emulsions stabilized by sweet potato protein under different pH levels combined with HPH have various applications in the food industry. HP treatment has also been shown to enhance the viscoelastic behavior of soy protein concentrate, contributing to improved protein content and textural properties of food products [89]. Interestingly, Teixeira et al. [102] demonstrated that pressurized liquid extraction (PLE), when used in conjunction with alkaline solvents (pH 9.0), effectively enhances protein recovery, yielding a high purity of 86.7% and improved thermal stability.

Moreover, HP treatment can effectively be combined with other modification techniques. Wang and Moraru [103] demonstrated optimal microstructural modification in milk protein concentrate (MPC) using pH 5.1 36 mg/g Ca combined with high-pressure homogenization (600 MPa, 5 °C, 3 min), achieving 58% porosity increase and 3.2-fold aggregate size enhancement. These findings highlight the potential use of HPH in the modification of MPC and open new avenues for the development of gel-based high-protein foods, such as puddings and portable protein bars. Figure 4 lists some applications of pH shifting combined with high pressure on protein.

These studies collectively highlight the effectiveness of HPH and pH shifting in improving the functional properties of plant proteins. By optimizing processing conditions such as pH and pressure levels, different functional modifications can be achieved for different protein sources [92]. Alternatively, variations in protein response to these treatments necessitate further optimization for each specific protein type. Although these physical treatments offer substantial benefits in enhancing protein functionality, practical applications must also consider economic feasibility, sensory attributes, and nutritional implications. Future research should further investigate the applicability of these methods in real food systems and assess their long-term effects on human health.

### 3.4. Other Physical Methods-Assisted pH Shift Treatments

In addition to the extensive research on combining pH shifting with physical methods such as heat treatment, ultrasound, and high pressure for protein processing, other physical techniques can also be integrated with pH shifting to modify proteins. Examples include pulsed electric field, microwaves, and cold ions. Although limited research has been conducted on combining these methods with pH shifting for protein modification, each technique has effectively been used in protein processing and modification, making them promising alternatives. The following section will discuss existing studies and potential applications of these methods in combination with pH shifting for protein modification.

#### 3.4.1. Pulsed Electric Field-Assisted pH Shifting

Pulsed electric field (PEF) is a non-thermal physical processing technology that applies a high-intensity electric field (typically 5–50 kV/cm) for a very short duration, inducing reversible or irreversible electroporation of biological cell membranes and thereby altering cell permeability [105]. This technology is widely used in food sterilization, ingredient extraction, and texture modification due to its efficiency, energy-saving properties, and ability to preserve food nutrients and sensorial characteristics.

Earlier studies show that PEF treatment (2.1 kV/cm) for 5–30 min at 25 °C significantly altered the secondary and tertiary structures of porcine myofibrillar proteins, and emulsification properties (EAI: 7.10 m^2^/g, ESI: 179.67%) [106]. By applying high-voltage pulses at 1.00–1.25 kV/cm electric field strength, 20 μs pulse width, and 50 Hz frequency, PEF induced partial unfolding of bovine muscle proteins by disrupting Z-disk and I-band junctions, exposing internal hydrolysis sites (e.g., troponin T and myosin heavy chain), thereby improving in vitro protein digestibility by 18–31% (*p* < 0.05) after 180 min of simulated gastro-small intestinal digestion. [107]. Compared with traditional thermal and high-pressure treatments, PEF can trigger conformational changes under milder conditions, facilitating protein digestion in the gastrointestinal tract [108]. Moreover, PEF improved protein hydrophilicity and dispersibility, while exposing additional reactive sites on amino acid residues, which facilitated interactions with other macromolecules such as polysaccharides and polyphenols [109]. Notably, the synergistic effects of pulsed electric field (PEF) are significantly enhanced when applied in combination with a pH shifting treatment. Under natural pH conditions (5.49–5.66) close to the isoelectric point of myofibrillar proteins, pH shifting initiates conformational loosening by altering protein charge distribution. When coupled with constant-current PEF (12–40 mA, 1000 Hz, needle-spring electrodes) or constant-voltage PEF (20 V/cm, 1000 Hz), the electric field further promotes protein unfolding and formation of smaller aggregates through suppression of disulfide bond formation and carbonyl compound accumulation (reducing oxidation by 18.3–28.7% compared to conventional thawing) [110]. This cooperative modification provides a structural basis for subsequent improvements in protein functionality.

Although comprehensive investigations into the structural effects of PEF and pH shifting remain limited, primarily due to the complex and dynamic nature of PEF-induced mechanisms, accumulating evidence continues to support its potential in modulating protein molecular conformation [111]. Building upon these structural alterations, the combined application of PEF and pH shifting has been shown to significantly improve key functional attributes of proteins. Specifically, this integrated strategy enhances solubility, reduces aggregation, and increases the exposure of reactive functional groups, thereby contributing to improved digestibility, emulsifying activity, foaming capacity, and other interfacial functionalities. For example, Wang et al. [112] reported that combined treatment with pH shifting (pH 11) and moderate PEF strength (10 kV/cm) synergistically increased the solubility of soy protein isolate from 26.06% to 70.34%. Corresponding enhancements in emulsifying activity and foaming ability were attributed to conformational unfolding, reduced particle size, and elevated surface hydrophobicity. These structural modifications optimized molecular flexibility and interfacial adsorption, thereby enhancing emulsifying and foaming behaviors. Similarly, the combination of PEF (1.4–1.8 kV/cm, 653–695 kJ/kg) and pH shifting, particularly at pH 4, effectively enhanced protein digestibility by improving the enzymatic hydrolysis of egg-white protein while reducing protein aggregation, thereby increasing its functionality [111]. Additionally, PEF pre-treatment (1.1 kV/cm, 1.53 kJ/kg) combined with pH shifting to 5.0 during protein precipitation significantly increased the extraction yield of grass-clover juice by 25% and crude protein release by 31% [113]. In summary, the combination of PEF and pH shifting offers a novel processing strategy for the functional enhancement of both plant and animal proteins, providing a theoretical and practical basis for the development of high-performance foods and nutritional delivery systems.

Recent advances suggest that the combination of PEF and pH shifting also holds considerable potential in the delivery of bioactive compounds, molecular encapsulation, selective recovery and processing of target biomolecules, improvement of protein product quality, and non-thermal microbial inactivation. In particular, this combined treatment has been shown to modulate transmembrane protein transport during membrane separation processes by altering charge distribution and the electrochemical potential gradient across both protein surfaces and membrane interfaces, thereby enhancing separation selectivity and increasing membrane flux [114]. For example, Wang et al. [115] successfully modified soy protein isolate using a combination of pulsed electric field (10 kV/cm) and pH shifting to pH 11 to develop SPI-based nanoparticles for efficient lutein encapsulation. In addition, PEF is also recognized as an effective processing technology which, when combined with appropriate pH adjustment, not only improves protein extraction yields but also reduces energy consumption, making it suitable for large-scale applications [116].

In conclusion, the use of PEF in plant protein processing and bioactive peptide preparation has gained increasing attention. However, the impact of PEF alone under neutral pH is relatively mild, and its ability to enhance purification efficiency remains limited. This is primarily due to the thermal dependency of PEF, and the partial structural protection of proteins under these conditions, which prevents excessive denaturation or degradation. To overcome these challenges, combining PEF with pH shifting has emerged as an effective strategy in protein processing.

#### 3.4.2. Microwave-Assisted pH Shifting

Microwave-assisted processing is a low-energy, low-pollution, non-thermal physical treatment technique that utilizes microwave radiation for rapid and uniform heating. Compared to conventional thermal methods, microwave treatment offers advantages such as faster heating rates, lower energy consumption, and improved uniformity. Because microwaves interact directly with polar groups within molecules, inducing molecular rotation and vibration, they thereby rapidly increase the internal temperature of the material and achieve uniform heating [117].

The combination of pH shifting and microwave treatment has been shown to significantly alter protein structure. Microwave irradiation promotes the rotation and vibration of polar moieties within protein molecules through rapid, directional heating, which induces unfolding and refolding of molecular structures within a short time frame. When coupled with pH shifting, the denaturation effects under extreme pH conditions are synergistically enhanced by microwave energy. This typically results in a decrease in *α*-helix content and an increase in *β*-sheet and random coil structures. Additionally, increases in surface hydrophobicity and free sulfhydryl content indicate molecular unfolding and the exposure of functional groups.

These conformational changes not only improve protein solubility and extraction efficiency but also enhance key functional properties such as emulsification, foaming, and antioxidant capacity. Consequently, this synergistic modification approach provides a promising strategy for the high-value utilization of proteins. pH shifting, as a widely used technique for improving protein solubility, when integrated with microwave treatment, has been demonstrated in numerous studies to synergistically enhance protein functionality. At the functional level, many studies have reported that the combination of microwave treatment and pH shifting significantly improved protein solubility, dispersibility, emulsifying properties, and thermal stability. For example, Han et al. [118] reported that the application of pH shifting (pH 12) and microwave heating (450 W, 90 °C, 2/4/6 min) to potato protein increased its solubility from 24.0 to 89.0%, surface hydrophobicity from 125 to 207, and reduced particle size from 249.95 μm to 90.37 nm, while simultaneously enhancing its dispersibility, emulsification, and thermal stability. Similarly, Das et al. [119] found that microwave-assisted extraction (600 W, 32 s) combined with pH modification (pH 8) optimized the yield and purity of soybean meal protein isolate, highlighting the synergistic effect of these techniques in enhancing extraction efficiency and quality control. Jahan et al. [120] further noticed that microwave (power 800, time 2 min) and pH shifting (pH 12) significantly improved the functional properties of mustard seed meal protein, thereby expanding its potential in plant protein processing.

In summary, the integration of microwave and pH shifting treatments not only induces substantial changes in protein structure but also effectively enhances their industrial application potential, particularly in the extraction, functional improvement, and sustainable utilization of plant-based proteins.

### 3.5. Multiple Methods-Assisted pH Shifting

With the advancement of protein processing research, the use of multi-method strategies for protein modification has emerged as a prominent area of focus. Modern processing approaches are no longer limited to the combination of two techniques; rather, when synergistic enhancements in performance and efficiency are observed, multiple processing technologies are increasingly applied in tandem. For proteins with inherently poor functional properties, specific and often complex processing conditions are required to achieve desirable modifications. Researchers have thus employed various techniques either to enhance specific functional attributes or to develop composite systems for targeted applications.

Building on this trend, an increasing number of studies have demonstrated the functional advantages of integrating physical, chemical, and enzymatic techniques to tailor protein properties for specific applications. Zhang et al. [121] optimized the extraction of proteins from herring by-products through the integration of radial discharge high-shear homogenization and ultrasound-assisted pH shifting treatment. The homogenization was conducted at 20,000 rpm for 3 min to disrupt cellular structures, followed by alkaline solubilization at pH 11.5 and subsequent isoelectric precipitation at pH 5.5. Ultrasound was applied at a power of 400 W for 10 min during the pH shifting process to enhance mass transfer and protein unfolding. This combined methodology not only improved protein recovery efficiency but also effectively retained the antioxidant activity of the extracts. These findings support the synergistic potential of mechanical and chemical treatments in modulating protein structure and functionality. Similarly, Igartúa et al. [18] investigated the structural and functional modification of rice protein isolates using a sequential treatment comprising pH shifting, high-intensity ultrasound, and thermal processing. The proteins were first solubilized under alkaline conditions at pH 11.0 to promote unfolding and dispersion. Subsequently, ultrasound treatment was applied using a probe-type sonicator (20 kHz) at full amplitude (100%) for 5 min to enhance mass transfer and facilitate molecular rearrangement. This was followed by thermal treatment at 85 °C for 30 min to stabilize the induced conformational changes. The combined treatment resulted in a marked increase in protein solubility and surface hydrophobicity, along with the formation of loosely associated aggregates. These structural alterations collectively contributed to improved plasticity and functional adaptability of the rice proteins. Furthermore, Sun et al. [37] examined the gelation behavior of soy protein isolate (SPI) subjected to enzymatic modification with transglutaminase (TGase) under controlled alkaline and thermal conditions. The treatment was performed at pH 8.0 with TGase added at a concentration of 20 U/g protein, followed by incubation at 50 °C for 60 min to facilitate cross-linking. This process resulted in a significant enhancement of gel hardness and water-holding capacity, which was attributed to the formation of covalent bonds between glutamine and lysine residues, thereby strengthening the protein network structure. Similarly, Karabulut et al. [122] explored the functional modification of hemp protein isolates through a combination of thermal ultrasound, HPH, and pH shifting. Alkaline solubilization was conducted at pH 11.0 for 60 min to promote protein unfolding, followed by isoelectric precipitation at pH 4.5. Ultrasound treatment was applied using a 25 kHz probe at 400 W for 30 min under controlled heating at 60 °C. Subsequently, the protein dispersion underwent HPH at 100 MPa for three cycles to induce further particle size reduction and structural disruption. This multi-step treatment significantly improved the solubility and emulsifying properties of hemp protein and contributed to a more relaxed and accessible protein structure, thereby enhancing its potential functionality in food formulations. Notably, the combination of multiple techniques is becoming a cutting-edge strategy in protein modification.

Like the protein-polysaccharide conjugates produced via pH shifting treatment described in Section 2, physical field-assisted pH regulation techniques can also promote covalent conjugation between proteins and polysaccharides. Jiang et al. found that ultrasound-assisted pH shifting accelerated the Maillard reaction of pea protein isolate and inulin and produced conjugates with a high degree of grafting [123]. Compared to the individual protein components, these conjugates exhibited significantly improved thermal stability, antioxidant activity, and interfacial properties. In a related study, Ding et al. [124] demonstrated that ultrasound-assisted pH shifting treatment effectively improved the interfacial adsorption behavior of whey protein isolate (WPI)-carboxymethyl cellulose (CMC) complexes. The treatment involved adjusting the system to an alkaline pH of 11.0 to induce protein unfolding, followed by acidification to pH 5.0 to facilitate complex formation near the isoelectric point of WPI. Ultrasound was applied at a power of 450 W and a frequency of 20 kHz for 10 min to enhance molecular interactions and structural rearrangements. This approach significantly increased the adsorption capacity at the oil–water interface, likely due to improved electrostatic and hydrogen bonding interactions between the protein and polysaccharide components. These findings underscore the potential of ultrasound-assisted pH shifting to modulate protein–polysaccharide interactions for emulsification applications.

These studies collectively demonstrate that the integration of pH shifting with physical treatments such as ultrasound, homogenization, and thermal processing can systematically optimize protein functionality by modulating conformational structure, aggregation behavior, and molecular interactions. This multi-technique exhibits synergistic effects, often yielding results greater than the sum of individual methods. However, despite the promising outcomes, the underlying mechanisms of such synergistic treatments remain complex. Current analyses rely primarily on macroscopic characterization, while the precise molecular mechanisms require further investigation, indicating a broad scope for future research.

## 4. Conclusions and Perspectives

Protein extraction and modification are critical to enhancing the value of agricultural products and advancing alternative protein sources. Among emerging strategies, the integration of pH shifting with physical treatments, such as thermal processing, ultrasound, high-pressure, and pulsed electric fields, has shown significant promise in improving protein structure and functionality. This combined approach enhances key functional properties, including solubility, emulsifying and foaming capacity, and thermal stability, while mitigating the limitations of single-modality treatments. Its application in emulsifiers, delivery systems, and functional foods underscores its potential for sustainable innovation in food science. Despite these advancements, several challenges hinder widespread industrial adoption. High equipment costs, particularly for high-pressure and pulsed electric field technologies, limit scalability. Additionally, extreme pH or intense physical treatments may cause partial protein degradation, emphasizing the need for optimized processing parameters. The synergistic mechanisms underlying these combined techniques remain insufficiently understood and warrant further molecular-level investigation. Moreover, comprehensive safety assessments and regulatory compliance evaluations are essential for ensuring consumer safety.

Future research should prioritize the development of cost-effective, environmentally sustainable, and efficient modification strategies that maintain protein integrity and functionality. Integrating insights from food science, biophysics, and computational modeling will be essential to optimize processes, clarify synergistic effects, and facilitate industrial translation. Addressing these challenges will enable the broader application of pH shifting combined with physical treatments, supporting the functional enhancement and sustainable utilization of protein resources in food and agricultural systems.

## Figures and Tables

**Figure 1 foods-14-02360-f001:**
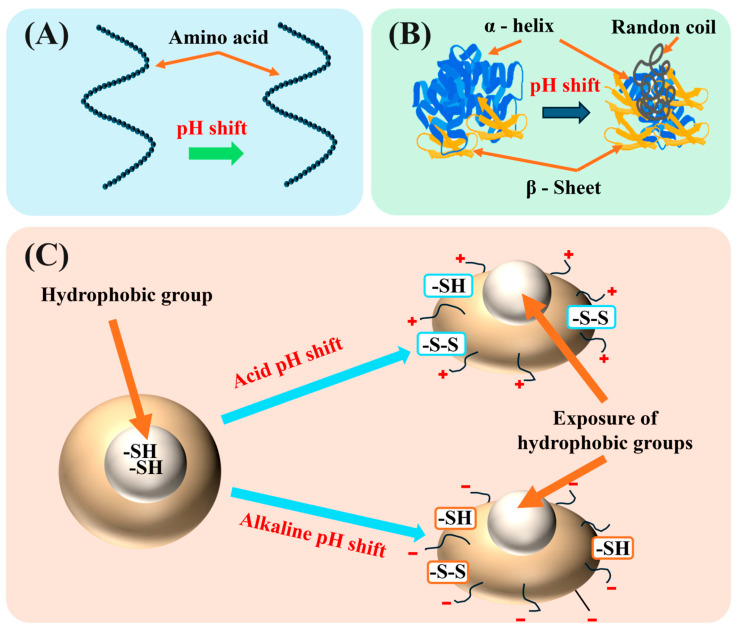
Effect of pH shifting on primary (**A**), secondary (**B**), and tertiary (**C**) structures of protein.

**Figure 2 foods-14-02360-f002:**
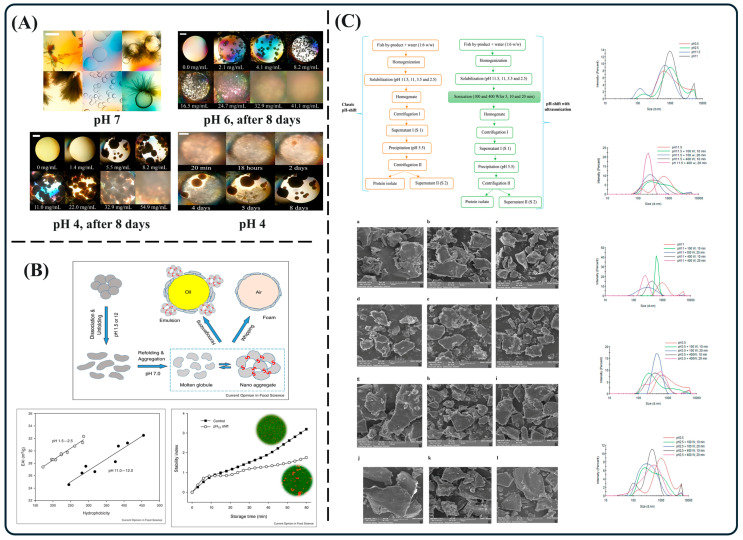
Phase behavior and spherulite growth of ovalbumin under different solution conditions (**A**) [30]; effects of pH changes on protein surface properties, emulsification performance, and emulsion stability (**B**) [32]; effects of ultrasound-assisted pH shift process on protein separation from fish by-products and its particle size distribution and surface microstructure (**C**) [33].

**Figure 3 foods-14-02360-f003:**
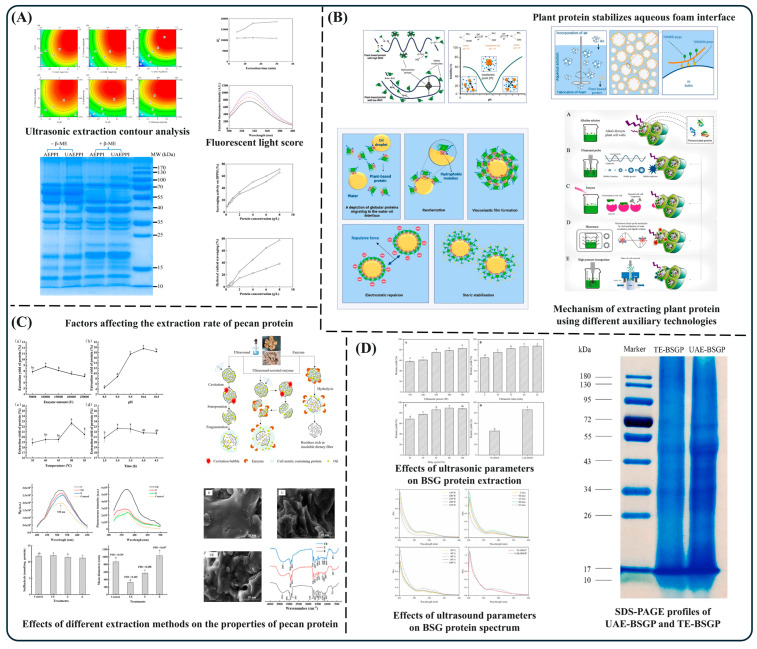
Effects of ultrasonic-assisted extraction on the properties and functions of pea protein (**A**) [11]; enhanced alkaline extraction techniques for plant-based proteins (**B**) [53]; effects of different extraction methods on the extraction rate, structure, and physicochemical properties of pecan protein (**C**) [54]; effects of ultrasonic parameters on BSG protein extraction yield, spectral characteristics, and molecular structure (**D**) [55].

**Figure 4 foods-14-02360-f004:**
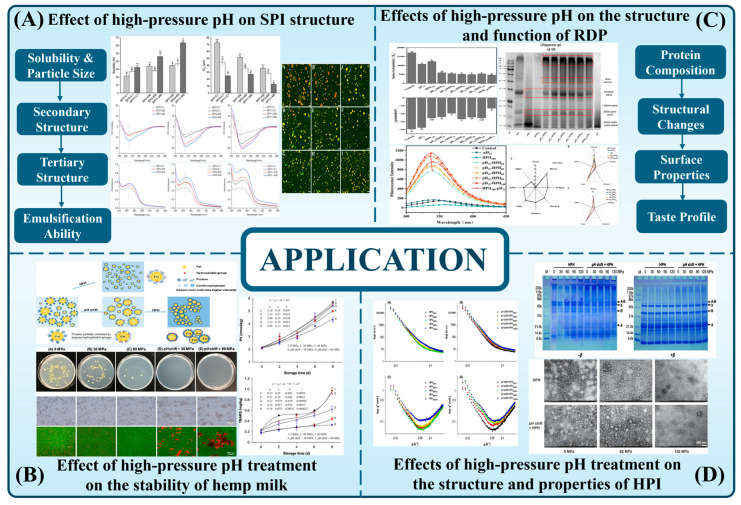
Effect of high pressure and pH treatment on the functionality and structure of various plant proteins. Soybean protein isolate (**A**) [96]; hemp milk (**B**) [104]; rice dreg protein (**C**) [85]; hempseed protein isolate (**D**) [92].

**Table 1 foods-14-02360-t001:** The effects of pH shifting combined with ultrasound on protein functionality.

Sample	Condition	Functional Improvements	Reference
Peanut protein	pH shifting (pH 12) combined with ultrasound (400 W, 20 kHz, 10 min, 25 °C)	The solubility was 111.4% higher than that of the control (d-ppi), respectively.	[68]
Soybean protein isolate (SPI), potato protein isolate (PPI4) and soybean/potato protein complex (SPI/PPI4 complex)	pH shifting (pH 12) combined with ultrasound (360 W, 20 kHz, 30 min, 25 °C)	Increased protein–water interactions, resulting in enhanced solubility. The gel properties were significantly improved after treatment, and the hardness, elasticity, and water-holding capacity (WHC) were significantly improved.	[65]
*Pleurotus ostreatus*	pH shifting (pH 12) combined with ultrasound (45 kHz, 64 min, 35 °C)	Enhanced emulsifying properties and foam stability of protein concentrate.	[62]
Yeast protein (YP)	pH shifting (pH 12, 2 h) combined with ultrasound (390 W, 25 kHz, 20 min, 25 °C)	The combined treatment greatly reduced the particle size of the protein and increased the solubility, foaming performance, and surface hydrophobicity (H_0_) value	[69]
Soy protein isolate (SPI)	Alkaline pH shifting (pH 9 1 h) combined with ultrasonic treatment (200 W, 300 W, 400 W, 10 min)	The viscosity of SPI decreased from 98.97 mPa. s to 22.83 mPa. s. These structural changes endow SPI with higher solubility (increasing from 81.13% to 91.53%), as well as better emulsifying and foaming properties.	[70]
Cottonseed meal protein (CSMP)	pH shifting (pH 1.5/3.5/9.5/11.5) and sonication (320 W, 20/60 kHz, 21 min)	Combined treatment at alkaline conditions increased absolute surface charge, solubility, foamability, and oil binding efficacy over control, and pH shifting alone (*p* < 0.05).	[60]
Pea protein isolate	pH shifting (pH 12, 1 h) and sonication (500 W, 20 kHz, 10 min, 25 °C)	PPI2 modified by ultrasound and pH change can successfully stabilize solid high internal phase lotion HIPEs with strong viscoelasticity and high stability.	[71]
Flaxseed protein isolate	pH shifting (pH 10, 2 h) and sonication (504 W, 20 kHz, 20 min, 25 °C)	UFPI-10 (FPI treated by ultrasound coupled with pH 10 cycling) possessed higher emulsification stability (ESI similar to 308.20 min), increasing by 1.74 times.	[72]
Chicken wooden breast myofibrillar protein (WBMP)	pH shifting (pH 11, 10 min, 4 °C) and sonication (400 W, 20 kHz)	WBMP emulsion more uniform, the gel strength and water-holding capacity of the protein gel increased.	[73]
Pine kernel protein (PKP)	Acidic pH shifting (pH 2, 30 min) and ultrasound (500 W, 20 kHz, 10 min, 30 °C)	Protein lotion shows higher viscoelasticity and stronger protein interaction, and lotion stability is enhanced.	[74]
Perilla protein isolate (PPI3)	pH shifting (pH 10/12, 30 min) and sonication (400 W, 20 kHz, 15 min)	the emulsifying and foaming properties of PPI3 could evidently enhance.	[66]
Chickpea protein isolate (CPI)	pH shifting (pH 2/12, 60 min) and sonication (300 W, 20 kHz, 20 min)	The foaming performance of CPI has significantly improved.	[75]
Coconut milk protein	pH shifting (pH 1/12, 60 min) and sonication (53 kHz, 40 W/L, 20 min, 25 °C)	Ultrasound can amplify the effect of pH shift on increasing the thermal stability of coconut milk by modifying functional properties and structures of coconut milk protein.	[76]
Shrimp proteins	alkaline pH shift (pH 12.5) combined with ultrasonication (300 W, 20 min)	The combined treatment of shrimp protein has a significant impact on its foaming and emulsifying properties.	[63]
Whey protein isolate	pH shifting (pH 2, 0–3 h) combined with ultrasound (600 W, 20 kHz, 30 min)	pH 2-shifting treatment combined with ultrasound could improve emulsion stability of WPI.	[77]
Native amaranth protein (APN)	pH shifting (pH 2/12, 1 h) combined with ultrasound (50% amplitude for 10 min)	Foaming capacity and stability were significantly increased in all treatments.	[78]
Faba bean protein isolate	pH shifting (pH 11) combined with ultrasound (1200 W, 20 kHz 10/20 min)	A striking enhancement in foaming capacity (from 93% to 306–386%) and stability (from 10 s to 473–974 s) was achieved by the combined treatment.	[79]
Barley protein isolate	pH shifting (pH 9) combined with ultrasound (20 kHz)	Ultrasound treatment improved both protein solubility and colloidal stability of the protein-enriched fraction at alkaline pH.	[80]

## Data Availability

No new data were created or analyzed in this study. Data sharing is not applicable to this article.

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
