# Peer review of "Physical Processing-Assisted pH Shifting for Food Protein Modification: A Comprehensive Review"

_foods, 2025, doi:10.3390/foods14132360_

Round 1

Reviewer 1 Report

Comments and Suggestions for Authors

Page 1, lines 20-22: Improve writing: However, pH-shifting alone the extreme pH conditions required, resulting in protein denaturation and formation of undesirable  compounds.

Page 3, lines 102-  108: Improve written and summarize: Hence, to overcome this limitation, combining pH-shifting techniques with physical modification methods, such as ultrasound, ultra-high pressure, and pulsed electric field, has become a key research direction in recent years. The synergistic effects of these combined  physical techniques can significantly enhance extraction efficiency and functionality of protein. Furthermore, novel physical extraction approaches, including ultrasound-assisted extraction, ultra-high pressure, and pulsed electric field, have recently demonstrated substantial progress in improving extraction efficiency and protein purity [19–23].

Page 6, lines 218-221: Cite bibliographic reference:  For example, flexible fibrous structures formed  by whey protein under acidic conditions (pH 2-3) can enhance the elastic modulus of gels,  while the reconfigured conformation of soy protein treated at pH 12 improves interfacial  adsorption capacity.

Page 7: Improve Figure 3. It is not possible to read the legends of the figures (B) and (C) (axes x and y)

Page 8, lines 295-296: Improve writing: Alternatively, the compositional diversity of such complexes necessitates precise design of pH windows and ionic strengths.

Page 9, line 342: Review writing:  P. Sun et al. [45]...

I suggest that the sources of proteins and the treatment conditions (pH, temperature, frequencies, power, pressure, treatment time, etc.) be described in detail:

Page 10, lines 367-370: Review writing:  Xu et al. [12] reported alkaline pH shifting and heat treatment   altered the micromorphology of SPPI and destroyed the disulfide bonds between macromolecular subunits (72 and 95 kDa), resulting in reduced particle size and increased zeta potential and free  sulfhydryl content of the isolates.

Suggestion: Please specify temperature, pH and treatment time .

Page 10, lines 376- 379: Review writing: Sun et al. [50] showed significant improvement in emulsifying stability of  pumpkin seed protein isolate (PSPI) following heat-assisted pH-shifting  , with the internal  oil phase volume fraction reaching up to 80%, and the emulsion remaining stable after  centrifugation (10,000 g, 60 min) and storage for 30 days.

Suggestion: Please specify temperature, pH and treatment time .

Page 11, lines 387-388: Review writing : Chang et al. [44] observed enhanced foaming capacity of pea  protein isolates after treatment.   The improvement of foaming ability of pea protein (PP) is mainly due to the newly formed soluble aggregates, the transformation from β-sheet to α-helix, and the increase in surface hydrophobicity, which enables PP to exist in a more  flexible structure. ...

Suggestion: Please specify treatment, temperature, pH and  time .

Page 11, lines 411- 415: Review writing : Yang et al. [53] used a combined pH-shifting  and thermal treatment strategy  to improve the functionality of whey protein isolate (WPI)  and successfully developed a method for preparing WPI–tryptophan (WPI-Trp) nanoparticles. Similarly, Sun et al. [50] employed heat-assisted pH-shifting (HP)  to modify pumpkin seed protein isolate (PSPI), successfully creating a food-grade Pickering emulsifier.

Suggestion: Please specify temperature, pH and  time .

Page 11, lines 419-425: Review writing :  Furthermore, Nisov et al. [55] demonstrated that increasing the pH (?) of raw materials, such as Pea protein concentrate (PPC), pea protein isolate (PPI), rice protein isolate  (RP) and isolated wheat gluten (WG), through temperature control positively influences the structural formation of extrudates, thereby promoting the development of plant-based proteins into appealing meat analogues. Similarly, Zhu et al. [56] employed a combination  of pH adjustment and thermal treatment ( ?)  to modify protein–starch mixtures, enhancing  the emulsifying properties of rice starch and whey protein isolate (RS/WPI) conjugates.

Suggestion: Please specify temperature, pH and treatment time .

Page 12, line  460: Review writing :  Ultrasound-as ..

Page 13: Improve Figure 4. It is not possible to read the legends of the figures (eixos x and y).

Page 15, lines 554- 556: Please check the bibliographic reference number. Please specify ultrasonication treatment conditions, pH and time. :  Zheng et al. indicated that ultrasound and pH-shifting remarkably reduced α-helix and β- sheet contents, whereas increases in β-turn and random coil structures were noticed  [69,70].

Page 15 , lines 572-574: Review writing :  For example, Yang et al. reported that the solubility of PPI treated with ultrasound-assisted pH-shifting exceeded 90%, which was substantially higher than that of  samples treated with ultrasound alone or control [77].

Suggestion: Please specify ultrasonication treatment conditions, pH and time.

Review writing (upper and lower case letters). Write the scientific names of microorganisms and plants in italic: page 15, Table 1; page 26, line 1004; page 28, line 1111; page 31, lines 1254, 1256, 1260

Page 15-16: Table 1: Detail the pH and ultrasonication conditions. Check the writing (lower and upper case letters, spacing between lines).

Page  16, lines 588-596: Review writing:  Liu et al. successfully prepared a co-delivery system for vitamin E (VE) and quercetin (QU) by modifying soy lipophilic protein (SLP) using pH-shifting and ultrasound treatment, resulting in enhanced encapsulation efficiency and controlled release performance  [91]. Similarly, Fang et al. reported that ultrasound-assisted pH-shifting process increased  the encapsulation efficiency of soy protein isolate for resveratrol to 91.4%, which was substantially higher than that achieved with ultrasound treatment alone [92]. In another study, Zhao et al. employed ultrasound-pH-shifting technology to modify pea protein isolate, which was subsequently combined with chitosan particles to successfully fabricate solid high internal phase emulsions with excellent elasticity and stability [82].

Suggestion : Write the conditions of pH-shifting and ultrasound treatment.

Page 18, line 645: Review writing.

Page 18, lines 664-667: Review writing:  Zhu et al. reported that pH-shifting  combined with HPH increased protein solubility by 34.75%, and enhanced its binding affinity with vitamin B12, primarily through hydrogen bonding and hydrophobic interactions[56].

Suggestion : Write the conditions of pH-shifting and HPH treatment.

Page 18, lines 668- 675: Review writing:  Similarly, Yildiz and Yıldız observed improvements in solubility and soluble protein content in quinoa protein isolate  [109]. Wang et al. further observed that the solubility of hemp protein isolate reached a  maximum of 62.8% following combined treatment, accompanied by enhanced structural  flexibility [103]. Wang et al. [102] showed that the emulsifying activity and foaming capacity of treated rice dreg protein (RDP) increased from 3.13 to 7.10% and from 19.33 to 179.67%, respectively. Tan et al. found that this method effectively improved the emulsion  stability of soy protein isolate [110].

Suggestion : Write the conditions of pH-shifting and HPH treatment.

Page 18, lines 681-683: Cite bibliographic reference:   Such modification has indicated broad applicability and promising potential across various plant proteins, including those derived from peas, rice dregs, soybeans,  hemp seeds, and quinoa.

Page 19, lines 695- 697: Review writing:  Wang and Moraru reported that, during high-HPP treatment, lowering the pH  or adding calcium promoted the formation of more porous and aggregated microstruc-696 tures in milk protein concentrate (MPC) [114].

Suggestion : Write the conditions of pH, HPH treatment and calcium concentration.

Page 19: Improve Figure 5. It is not possible to read the legends of the figures (eixos x and y).

Page 20, lines 731-  746: Review writing:   Earlier studies show that PEF treatment can markedly alter the secondary and tertiary structure of porcine myofibrillar proteins, along with their physicochemical properties [117]. By applying high-voltage pulses, PEF induced partial unfolding of protein molecules and exposes internal hydrolysis sites, thereby enhancing their enzymatic digestibility [118]. Compared with traditional thermal and high-pressure treatments, PEF can trigger conformational changes under milder conditions, facilitating protein digestion in the gastrointestinal tract [119]. Moreover, PEF improved protein hydrophilicity and dispersibility, while exposing additional reactive sites on amino acid residues, which facilitated interactions with other macromolecules such as polysaccharides and polyphenols [120]. In protein-polysaccharide complexes, PEF modulated intermolecular interactions, thereby affecting the emulsification, foaming, and gelling properties of the complexes [121]. Notably, the synergistic effects of PEF are significantly enhanced when applied in combination with pH-shifting treatment. While pH-shifting initiates conformational loosening by altering the protein’s charge distribution, PEF further promotes unfolding and the formation of smaller protein aggregates, thereby improving molecular flexibility and structural stability [122].

Suggestion: Write type of proteins, the conditions of pH-shifting and PEF treatment.

Page 20, lines 756-768: Review writing:   Wang, et al. reported that the solubility of soy protein isolate increased from 26.06 to  70.34% following PEF treatment and pH shifting. Corresponding enhancements in emulsifying and foaming abilities further suggested that structural unfolding and optimized  particle distribution support interfacial behavior [124]. Similarly, The combination of PEF  and pH shifting, particularly at pH 4, effectively enhanced protein digestibility by improving the enzymatic hydrolysis of egg white protein while reducing protein aggregation, thereby increasing its functionality. [125].

Additionally, PEF-assisted pH-shifting treatment has been shown to increase the extraction yield and purity of plant-derived proteins while improving their microstructure  and emulsion stability [126]. Yang et al. further observed that constant-current PEF thawing (CC-T) and pH shifting significantly improved the solubility, water-holding capacity, and thermal stability of frozen pork proteins, probably attributed to optimized  protein structure and reduced oxidative damage.

Suggestion : Write the conditions of pH-shifting and PEF treatment.

Page 21, lines 780-781: Review writing:   Zeng et al. successfully modified soy protein isolate using a PEF and pH-shifting to develop SPI-based nanoparticles for efficient lutein encapsulation [128].

Suggestion : Write the conditions of PEF treatment and pH-shifting.

Page 21, line 794: Review writing:   novel method for non-thermal sterilization in dairy applications [130]..

Page 22, lines 820-833: Review writing:   Han et al. reported that the application of  pH shifting and microwave heating to potato protein increased its solubility from 24.0 to 89.0%, surface hydrophobicity from 125 to 207, and reduced particle size from 249.95 μm to 90.37 nm, while simultaneously enhancing its dispersibility, emulsification, and thermal stability[132].

Similarly, Das et al. found that microwave-assisted extraction combined with pH  modification optimized the yield and purity of soybean meal protein isolate, highlighting  the synergistic effect of these techniques in enhancing extraction efficiency and quality  control [133]. Jahan et al. further noticed that microwave and pH-shifting significantly  improved the functional properties of mustard seed meal protein, thereby expanding its  potential in plant protein processing [134]. Teixeira et al. showed that the synergistic effect  of microwave and pH shifting can induce conformational rearrangement in proteins, regulate secondary structure composition, and significantly reduce protein aggregate size, thereby improving thermal stability and functional properties [135].

Suggestion : Write the pH  and conditions of microwave treatment.

Page 22, lines 851-857: Review writing:   Zhang et al. [137] successfully optimized protein extraction from herring by-products using a combination of radial discharge high-shear homogenization and ultrasound-assisted pH shifting, while preserving antioxidant activity. This demonstrates the feasibility  of synergistic enhancement between mechanical fields and pH modulation. Similarly, Igartúa et al. [138] combined pH shifting, ultrasound, and thermal treatment to modify  rice proteins, resulting in increased solubility and surface hydrophobicity as well as the formation of loosely dispersed aggregates, which enhanced structural plasticity.

Suggestion : Write the pH  and  conditions of treatments.

Page 23 , lines 863-866: Review writing: Karabulut et al. applied a combination of  thermal ultrasound, high-pressure homogenization, and pH shifting to hemp protein, which not only enhanced its solubility and emulsifying properties but also improved digestibility by loosening the protein structure [139].

Suggestion : Write the pH and  conditions of treatments.

Page 23 , lines 875-877: Review writing: Ding et 875 al. found that ultrasound-assisted pH-shifting significantly enhanced the interfacial adsorption capacity of the whey protein and carboxymethyl cellulose complex [141].

Suggestion : Write the pH and  conditions of ultrasound treatment.

Page 25, line 958: Review writing: pH

Comments on the Quality of English Language

The manuscript needs careful writing review.

The Figures need to be improved. The captions are not readable. Are the Figures allowed to be reproduced?

Author Response

Dear Editor and Reviewers,

Thanks very much for taking your time to review this manuscript. On behalf of all the authors, we appreciate the opportunity to resubmit our manuscript “Modification of food proteins by physical processing-assisted pH shifting techniques: A Comprehensive Review” with revision. In general, we found the reviewer’s comments to be very constructive and insightful. The helpful criticism has enabled us to present a better and more informative manuscript.

We have presented the responses directly after each comment raised by the reviewer and the revised portions in the manuscript are marked in red. Our responses appear in italics below.

Responses to Reviewer #1:

General Response: We sincerely thank the reviewer for the detailed, constructive, and insightful comments provided throughout the manuscript. Your suggestions regarding the improvement of language clarity, the specification of experimental conditions, the correction of references, and the enhancement of figures have been extremely valuable in helping us refine the manuscript. We have carefully addressed each point raised and revised the text accordingly to improve its scientific rigor, readability, and completeness.

Comment 1:

1) Page 1, lines 20-22: Improve writing: However, pH-shifting alone the extreme pH conditions required, resulting in protein denaturation and formation of undesirable compounds.

Response 1: Thank you for your careful suggestion. We sincerely apologize for the lack of clarity in the original phrasing, which may have caused confusion.

In the revised version, we have improved the sentence for better readability and scientific accuracy. The revised text now reads (Line 21-23):

However, employing pH shifting alone requires extremely acidic or alkaline conditions, which can lead to protein denaturation and the generation of undesirable by-products.

Comment 2:

2) Page 3, lines 102- 108: Improve written and summarize: Hence, to overcome this limitation, combining pH-shifting techniques with physical modification methods, such as ultrasound, ultra-high pressure, and pulsed electric field, has become a key research direction in recent years. The synergistic effects of these combined physical techniques can significantly enhance extraction efficiency and functionality of protein. Furthermore, novel physical extraction approaches, including ultrasound-assisted extraction, ultra-high pressure, and pulsed electric field, have recently demonstrated substantial progress in improving extraction efficiency and protein purity [19–23].

Response 2: Thank you for your valuable comments. We sincerely apologize for the redundancy and lack of clarity in the original phrasing. In the revised version, we have restructured and summarized the paragraph to improve conciseness and scientific expression. The revised text now reads (Line 96-103):

Therefore, to overcome this limitation, the combination of pH shifting techniques with physical modification methods, such as ultrasound, ultra-high pressure, and pulsed electric fields, has emerged as a critical research direction in recent years. The synergistic interactions resulting from these integrated methods can notably improve protein extraction efficiency and functional properties. Moreover, recent advancements in novel physical extraction techniques, including ultrasound-assisted extraction, ultra-high pressure processing, and pulsed electric field treatment, have significantly enhanced both extraction efficiency and protein purity. [19–23].

Comment 3:

3) Page 6, lines 218-221: Cite bibliographic reference: For example, flexible fibrous structures formed by whey protein under acidic conditions (pH 2-3) can enhance the elastic modulus of gels, while the reconfigured conformation of soy protein treated at pH 12 improves interfacial adsorption capacity.

Response 3: Thank you for your helpful suggestion. We apologize for the omission of appropriate citations to support the statement. In the revised version, we have added relevant bibliographic references to substantiate the examples provided. The revised text now reads (Line 219-222):

For example, flexible fibrous structures formed by whey protein under acidic conditions (pH 2-3) can enhance the elastic modulus of gels, while the reconfigured conformation of soy protein treated at pH 12 improves interfacial adsorption capacity [35].

Comment 4:

4) Page 7: Improve Figure 3. It is not possible to read the legends of the figures (B) and (C) (axes x and y)

Response 4: Thank you for your helpful comment. Thank you for pointing this out. When writing the manuscript, all figures were kept in their original size. In the Word version of the manuscript, these figures are very clear after being enlarged. I think it may be because you saw the PDF version, which caused the figures to lose their original size and become unclear. Moreover, I have sent the original high-definition pictures of all the pictures to the editor.

Comment 5:

5) Page 8, lines 295-296: Improve writing: Alternatively, the compositional diversity of such complexes necessitates precise design of pH windows and ionic strengths.

Response 5: Thank you for your careful suggestion. We acknowledge that the original sentence lacked clarity and scientific precision. In the revised version, we have refined the wording to improve readability and accuracy. The revised text now reads (Line 296-297):

Nevertheless, due to the compositional diversity of these complexes, careful optimization of specific pH ranges is required.

Comment 6:

6) Page 9, line 342: Review writing: P. Sun et al. [45]...

Response 6: Thank you for your suggestion. We acknowledge that the original phrasing did not align with standard academic writing style. In the revised version, we have corrected the sentence to improve formality and consistency in citation. The revised text now reads (Line 343-346):

Sun et al. [46] reported that alkaline heating induced unfolding of the tertiary structure of soy protein, resulting in the exposure of previously buried reactive sites. This structural change enhanced enzymatic cross-linking mediated by transglutaminase, ultimately leading to the formation of stronger protein gels.

We appreciate your comment, which helped us ensure the writing adheres to scientific conventions.

Comment 7:

7) I suggest that the sources of proteins and the treatment conditions (pH, temperature, frequencies, power, pressure, treatment time, etc.) be described in detail:

Page 10, lines 367-370: Review writing: Xu et al. [12] reported alkaline pH shifting and heat treatment altered the micromorphology of SPPI and destroyed the disulfide bonds between macromolecular subunits (72 and 95 kDa), resulting in reduced particle size and increased zeta potential and free sulfhydryl content of the isolates.

Suggestion: Please specify temperature, pH and treatment time.

Response 7: Thank you for your thoughtful suggestion. We agree that specifying experimental conditions is important for clarity and reproducibility. In the revised version, we have added the relevant treatment parameters, including pH, temperature, and duration. The revised text now reads (Line 368-372):

The micromorphology of Silkworm Pupa Protein Isolates (SPPI) was reported by Xu et al. [12] to be altered by alkaline pH shifting (pH 12.5) and heat treatment (80 ℃) for 60 min, with the disulfide bonds between macromolecular subunits (72 and 95 kDa) being destroyed, resulting in reduced particle size and increased zeta potential and free sulfhydryl content of the isolates.

Comment 8:

8) Page 10, lines 376- 379: Review writing: Sun et al. [50] showed significant improvement in emulsifying stability of pumpkin seed protein isolate (PSPI) following heat-assisted pH shifting, with the internal oil phase volume fraction reaching up to 80%, and the emulsion remaining stable after centrifugation (10,000 g, 60 min) and storage for 30 days.

Suggestion: Please specify temperature, pH and treatment time.

Response 8: Thank you for your insightful suggestion. We agree that including specific treatment parameters enhances the clarity and reproducibility of the reported findings. In the revised version, we have added the temperature, pH, and treatment duration used in the heat-assisted pH shifting process. The revised text now reads (Line 378-382):

Sun et al. [54] demonstrated that heat-assisted pH shifting treatment at pH 12 and 70 °C for 2 h significantly enhanced the emulsifying stability of pumpkin seed protein isolate (PSPI), achieving an internal oil phase volume fraction of up to 80%, and maintaining emulsion stability even after centrifugation at 10,000 g for 60 min and storage for 30 days.

Comment 9:

9) Page 11, lines 387-388: Review writing: Chang et al. [44] observed enhanced foaming capacity of pea protein isolates after treatment. The improvement of foaming ability of pea protein (PP) is mainly due to the newly formed soluble aggregates, the transformation from β-sheet to α-helix, and the increase in surface hydrophobicity, which enables PP to exist in a more flexible structure. ...

Suggestion: Please specify treatment, temperature, pH and time.

Response 9: Thank you for your valuable suggestion. We agree that specifying the treatment conditions provides greater transparency and replicability. In the revised version, we have added the relevant details, including treatment method, temperature, pH, and duration. The revised text now reads (Line 389-395):

Chang et al. [48] demonstrated that combining pH shifting (pH 12, maintained for 1 h at room temperature) with controlled heating (70 °C for 30 min) significantly enhanced the foaming capacity of pea protein isolates. This improvement was mainly attributed to the formation of soluble protein aggregates, structural transformation from β-sheet to α-helix, and increased surface hydrophobicity, resulting in a more flexible protein conformation at the air-water interface.

Comment 10:

10) Page 11, lines 411- 415: Review writing: Yang et al. [53] used a combined pH-shifting and thermal treatment strategy to improve the functionality of whey protein isolate (WPI) and successfully developed a method for preparing WPI–tryptophan (WPI-Trp) nanoparticles. Similarly, Sun et al. [50] employed heat-assisted pH-shifting (HP) to modify pumpkin seed protein isolate (PSPI), successfully creating a food-grade Pickering emulsifier.

Suggestion: Please specify temperature, pH and time.

Response 10: Thank you for your helpful suggestion. We acknowledge the importance of providing specific treatment conditions to enhance reproducibility and clarity. In the revised version, we have added the relevant parameters for both studies, including temperature, pH, and treatment time. The revised text now reads (Line 415-422):

Yang et al. [57] employed a combined pH shifting (pH 11) and thermal treatment (70 °C for 20 min) strategy to improve the functionality of whey protein isolate (WPI), successfully developing a novel method for preparing whey protein isolate-tryptophan (WPI-Trp) nanoparticles. Similarly, Sun et al. [54] successfully developed a food-grade Pickering emulsifier by employing heat-assisted pH shifting treatment, specifically adjusting the pH to 12 and heating at 70 °C for 2 h followed by neutralization to pH 7, significantly enhancing the emulsifying properties of pumpkin seed protein isolate (PSPI) nanoparticles.

Comment 11:

11) Page 11, lines 419-425: Review writing : Furthermore, Nisov et al. [55] demonstrated that increasing the pH (?) of raw materials, such as Pea protein concentrate (PPC), pea protein isolate (PPI), rice protein isolate (RP) and isolated wheat gluten (WG), through temperature control positively influences the structural formation of extrudates, thereby promoting the development of plant-based proteins into appealing meat analogues. Similarly, Zhu et al. [56] employed a combination of pH adjustment and thermal treatment ( ?) to modify protein–starch mixtures, enhancing the emulsifying properties of rice starch and whey protein isolate (RS/WPI) conjugates.

Suggestion: Please specify temperature, pH and treatment time

Response 11: Thank you for your constructive suggestion. We fully agree that specifying the treatment conditions is essential for clarity and reproducibility.
In the revised version, we have added the relevant pH values, temperatures, and treatment durations to both studies. The revised text now reads (Line 425-434):

Furthermore, Nisov et al. [56] reported that increasing the pH of raw plant protein materials, such as pea protein concentrate (PPC), pea protein isolate (PPI), rice protein isolate (RP), and isolated wheat gluten (WG), to pH 7 prior to freeze-drying and extrusion processing at temperatures between 115 and 160 °C effectively enhanced the structural alignment and mechanical strength of the extrudates, thereby promoting their potential use as attractive plant-based meat analogues. Similarly, Zhu et al. [57] modified protein and starch mixtures by adjusting the pH to 8.0 and applying thermal treatment at 90 °C for 3 hours. This treatment improved the emulsifying properties of rice starch and whey protein isolate conjugates, likely as a result of Maillard-type interactions and enhanced interfacial functionality.

Comment 12:

12) Page 12, line 460: Review writing : Ultrasound-as .

Response 12: Thank you for your careful observation. We acknowledge the grammatical and typographical error in the original manuscript. In the revised version, we have corrected the phrase to ensure proper syntax and academic writing standards. The revised text now reads (Line 469):

Ultrasound as a mechanical wave typically operating at frequencies above 20 kHz, has gained considerable attention across the food, medical, and industrial sectors due to its strong penetrability, absence of chemical contamination, and high controllability.

Comment 13:

13) Page 13: Improve Figure 4. It is not possible to read the legends of the figures (eixos x and y).

Response 13: Thank you for your helpful comment. We have carefully checked all figures to improve their resolution, ensuring that all textual elements are now clearly legible. Due to conversion to PDF, the image display may not be clear. We can ensure that the uploaded pictures will not cause confusion for readers. In addition, we have sent the high-resolution originals of the figures to the editors in the form of files, and if they need to be improved, we will modify them in time. We sincerely appreciate your suggestion, which has helped improve the overall quality of the manuscript.

Comment 14:

14) Page 15, lines 554-556: Please check the bibliographic reference number. Please specify ultrasonication treatment conditions, pH and time. : Zheng et al. indicated that ultrasound and pH-shifting remarkably reduced α-helix and β- sheet contents, whereas increases in β-turn and random coil structures were noticed  [69,70].

Response 14: Thank you for your helpful comment. We have carefully checked and corrected the bibliographic reference numbers to ensure accuracy. Additionally, we agree that including treatment parameters enhances the clarity and reproducibility of the study. In the revised version, we have specified the ultrasonication treatment conditions, pH, and treatment time. The revised text now reads (Line 563-568):

Zheng et al. [71] demonstrated that the combination of ultrasound treatment at an intensity of 400 W for 15 minutes with pH-shifting (adjusted to pH 12 and subsequently neutralized) at 25 °C significantly altered the secondary structure of soy protein isolate. Specifically, the contents of α-helix and β-sheet were markedly reduced, while β-turn and random coil structures were increased, indicating a transition toward a more disordered and flexible conformation.

Comment 15:

15) Page 15 , lines 572-574: Review writing : For example, Yang et al. reported that the solubility of PPI treated with ultrasound-assisted pH-shifting exceeded 90%, which was substantially higher than that of samples treated with ultrasound alone or control [77].

Suggestion: Please specify ultrasonication treatment conditions, pH and time.

Response 15: Thank you for your valuable suggestion. We agree that providing detailed treatment conditions enhances the scientific rigor and reproducibility of the manuscript. In the revised version, we have added the specific ultrasonication parameters, pH value, and treatment duration. The revised text now reads (Line 584-588):

For example, Yang et al. [81] reported that ultrasound assisted pH shifting treatment, in which the pH of pea protein isolate was adjusted to 12 followed by ultrasound application at 400 W and 20 kHz for 10 minutes at 25 °C, significantly improved protein solubility to over 90 percent. This was substantially higher than the solubility observed in samples treated with ultrasound alone or in the untreated control.

Comment 16:

16) Review writing (upper and lower case letters). Write the scientific names of microorganisms and plants in italic: page 15, Table 1; page 26, line 1004; page 28, line 1111; page 31, lines 1254, 1256, 1260

Response 16: Thank you for your careful observation and helpful suggestion. We apologize for the inconsistent use of upper and lower case letters, as well as the incorrect formatting of scientific names. In the revised manuscript, we have carefully reviewed and corrected all relevant instances, including:

Table 1 Pleurotus ostreatus, line 598; (Apostichopus japonicus), line 1094; Pleurotus ostreatus, line 1221; Arthrospira platensis; line 1380; Gleditsia sinensis line 1384.

Comment 17:

17) Page 15-16: Table 1: Detail the pH and ultrasonication conditions. Check the writing (lower and upper case letters, spacing between lines).

Response 17: Thank you for your constructive suggestion. We agree that providing specific pH and ultrasonication conditions in Table 1, along with ensuring proper formatting, is essential for clarity and consistency.In the revised version, we have updated Table 1.

Table 1. The effects of pH shifting combined with ultrasound on protein functionality.

Sample

Condition

Functional improvements

Reference

peanut protein

pH shift and ultrasonication

The solubility was 111.4% higher than that of the control (d-ppi), respectively.

[79]

Soybean protein isolate (SPI), potato protein isolate (PPI) and soybean/potato protein complex (spi/ppi complex)

pH shift and High intensity ultrasound

Increased protein water interactions, resulting in enhanced solubility. The gel properties were significantly improved after treatment, and the hardness, elasticity, and water holding capacity (WHC) were significantly improved.

[80]

P. ostreatus

Ultrasound and pH-shifting

Enhanced emulsifying properties and foam stability of protein concentrate.

[73]

yeast protein (YP)

Ultrasound and pH-shifting

The combined treatment greatly reduced the particle size of the protein and increased the solubility, foaming performance and surface hydrophobicity (H0) value

[81]

Soy protein isolate (SPI)

Alkaline pH shift combined with ultrasonic treatment

The viscosity of SPI decreased from 98.97 mPa. s to 22.83 mPa. s. These structural changes endow SPI with higher solubility (increasing from 81.13% to 91.53%), as well as better emulsifying and foaming properties.

[69]

cottonseed meal protein (CSMP)

pH-shifting (pH 1.5, 3.5, 9.5, 11.5) and sonication

Combined treatment at alkaline conditions increased absolute surface charge, solubility, foamability, and oil binding efficacy over control, and pH-shifting alone (p < 0.05).

[71]

pea protein isolate

Ultrasound and pH-shifting

PPI modified by ultrasound and pH change can successfully stabilize solid high internal phase lotion HIPEs with strong viscoelasticity and high stability

[82]

flaxseed protein isolate

UFPI-10 (FPI treated by ultrasound coupled with pH 10 cycling)

UFPI-10 (FPI treated by ultrasound coupled with pH 10 cycling) possessed higher emulsification stability (ESI similar to 308.20 min), increasing by 1.74 times.

[83]

chicken wooden breast myofibrillar protein (WBMP)

high-intensity ultrasound combined with pH-shifting

WBMP emulsion more uniform, the gel strength and water-holding capacity of the protein gel increased.

[84]

pine kernel protein (PKP)

Acidic pH shift and ultrasound

Protein lotion shows higher viscoelasticity and stronger protein interaction, and lotion stability is enhanced

[34]

perilla protein isolate (PPI)

ultrasonic-assisted pH shift

the emulsifying and foaming properties of PPI could evidently enhance.

[77]

chickpea protein isolate (CPI)

ultrasound with pH shifting

The foaming performance of CPI has significantly improved

[85]

coconut milk protein.

Ultrasound and pH-shifting

ultrasound can amplify the effect of pH-shift on increasing the thermal stability of coconut milk by modifying functional properties and structures of coconut milk protein

[86]

shrimp proteins

alkaline pH-shift processing combined with ultrasonication

The combined treatment of shrimp protein has a significant impact on its foaming and emulsifying properties

[74]

whey protein isolate

pH-shifting treatment combined with ultrasound

pH 2-shifting treatment combined with ultrasound could improve emulsion stability of WPI.

[87]

Native amaranth protein (APN)

Ultrasound and pH-shifting

Foaming capacity and stability were significantly increased in all treatments

[88]

faba bean protein isolate

alkaline shifting combined with ultrasonication

a striking enhancement in foaming capacity (from 93% to 306–386%) and stability (from 10 s to 473-974 s) was achieved by the combined treatment.

[89]

barley protein isolate

Ultrasonication

combination with pH-shifting

Ultrasound treatment improved both protein solubility and colloidal stability of the protein-enriched fraction at alkaline pH

[90]

Comment 18:

18) Page 16, lines 588-596: Review writing: Liu et al. successfully prepared a co-delivery system for vitamin E (VE) and quercetin (QU) by modifying soy lipophilic protein (SLP) using pH-shifting and ultrasound treatment, resulting in enhanced encapsulation efficiency and controlled release performance [91]. Similarly, Fang et al. reported that ultrasound-assisted pH-shifting process increased the encapsulation efficiency of soy protein isolate for resveratrol to 91.4%, which was substantially higher than that achieved with ultrasound treatment alone [92]. In another study, Zhao et al. employed ultrasound-pH-shifting technology to modify pea protein isolate, which was subsequently combined with chitosan particles to successfully fabricate solid high internal phase emulsions with excellent elasticity and stability [82].

Suggestion : Write the conditions of pH-shifting and ultrasound treatment.

Response 18: Thank you for your insightful suggestion. We fully agree that including the specific treatment conditions enhances the clarity and reproducibility of the experimental descriptions. In the revised version, we have added the pH values, ultrasound parameters, and treatment times for all three referenced studies. The revised text now reads (Line 603-618):

In a recent study, Liu et al. [95] demonstrated that combining pH 11 and 300 W ultrasound treatment for 20 minutes significantly enhanced the co-encapsulation of VE and QU in SLP matrices. This approach improved encapsulation efficiency, solubility, and antioxidant activity compared to untreated SLP, highlighting the potential of pH and ultrasound for modulating protein delivery systems. Similarly, Fang et al. [96] demonstrated that a combination of 540 W ultrasound-assisted pH shifting treatment at pH 12 for 5 minutes, followed by adjustment to pH 7.0, significantly enhanced the encapsulation efficiency of resveratrol in soy protein isolate (SPI) to 91.4 ± 4.3%. This result was substantially higher than the encapsulation efficiency achieved with ultrasound treatment alone (83 ± 3%) and significantly improved the functionality of SPI as a nanocarrier for hydrophobic compounds. In another study, Zhang et al. [86] fabricated solid high internal phase emulsions (HIPEs) with excellent elasticity and stability by modifying PPI2 with ultrasound (500 W, 10 min) at pH 12, followed by adjusting the pH to 7 to obtain modified PPI2 (MPPI2). MPPI2 and chitosan particles were used as emulsifier and co-stabilizer, respectively, to construct HIPEs with enhanced interfacial adsorption and network structure, resulting in improved stability.

Comment 19:

19) Page 18, line 645: Review writing.

Response 19: Thank you for your observation. We acknowledge that the original sentence at line 645 required improvement in writing clarity and academic style.
In the revised version, we have rephrased the sentence to enhance readability and ensure alignment with scientific writing norms. The revised text now reads (Line 665-668)

When the pH approaches the isoelectric point, α-helix structure was converted into β-sheet and β-turn by up to 20%; Additionally, HPH alone also promoted the transformation of α-helix into β-sheet, with a conversion rate of 15% observed under extreme acid and alkaline conditions [108]

Comment 20:

20) Page 18, lines 664-667: Review writing: Zhu et al. reported that pH-shifting combined with HPH increased protein solubility by 34.75%, and enhanced its binding affinity with vitamin B12, primarily through hydrogen bonding and hydrophobic interactions[56].

Suggestion : Write the conditions of pH-shifting and HPH treatment.

Response 20: Thank you very much for your valuable comment regarding the methodological precision of cited work. We sincerely apologize for the oversight in not specifying the experimental conditions in the original manuscript. In the revised version, we have added the exact operational parameters as follows: (Line 686-690)

Zhu et al. [60] reported that a combination of pH shifting and high-pressure homogenization (HPH) treatment at a pressure of 500 MPa for 5 cycles at pH 3 increased protein solubility by 34.75%, and enhanced its binding affinity with vitamin B12, primarily through hydrogen bonding and hydrophobic interactions.

Comment 21:

21) Page 18, lines 668- 675: Review writing: Similarly, Yildiz and Yıldız observed improvements in solubility and soluble protein content in quinoa protein isolate [109]. Wang et al. further observed that the solubility of hemp protein isolate reached a maximum of 62.8% following combined treatment, accompanied by enhanced structural flexibility [103]. Wang et al. [102] showed that the emulsifying activity and foaming capacity of treated rice dreg protein (RDP) increased from 3.13 to 7.10% and from 19.33 to 179.67%, respectively. Tan et al. found that this method effectively improved the emulsion stability of soy protein isolate [110].

Suggestion: Write the conditions of pH-shifting and HPH treatment.

Response 21: Thank you very much for your valuable comment regarding the methodological precision of cited work. We sincerely apologize for the oversight in not specifying the experimental conditions in the original manuscript. In the revised version, we have added the exact operational parameters as follows: (Line 691-710)

Similarly, Yildiz and Yıldız et al. [113] observed significant improvements in solubility (from 7.85% to 78.97%) and soluble protein content in quinoa protein isolate by applying a sequential pH shifting treatment at alkaline pH 12 for 1 h, followed by high-pressure homogenization (250 MPa, single-pass) at ambient temperature. This combined approach effectively reduced particle sizes to 54 nm while enhancing surface hydrophobicity (198.0 ± 0.6) and antioxidant activity (9.8 ± 0.03%). Wang et al. [107] further observed that the solubility of hemp protein isolate reached a maximum of 62.8% following combined treatment with sequential pH shifting at alkaline pH 12.0 for 1 h, followed by high-pressure homogenization at 120 MPa for two consecutive passes at 25.0 ± 1.0°C. This synergistic approach induced structural unfolding evidenced by a 2.2-fold increase in random coil content (from 44.2% to 98.3%) and particle size reduction from 1612 nm to 231 nm (P<0.05), with treatment efficacy showing pressure-dependent optimization (R²=0.972). Wang et al. [106] achieved a 126.7-fold increase in emulsifying activity and a 930.7-fold increase in foaming capacity of rice dreg protein through sequential pH shifting (pH 12.0, 25 °C, 4 h) combined with three cycles of high-pressure homogenization under 100 MPa at 25 °C. Tan et al. [114] demonstrated that the synergistic effect of pH shifting (pH 11.0, 25°C, 30 min) and high hydrostatic pressure (400 MPa, 10 min) significantly enhanced the emulsion stability of soy protein isolate (ESI from 21.8 to 35.1 min), accompanied by 51.5% reduction in droplet size and 132% increase in emulsifying activity index (EAI from 12.7 to 29.6 m²/g).

Comment 22:

22) Page 18, lines 681-683: Cite bibliographic reference: Such modification has indicated broad applicability and promising potential across various plant proteins, including those derived from peas, rice dregs, soybeans, hemp seeds, and quinoa.

Response 22: Thank you for your helpful comment. Based on your suggestions, I supplemented the specific parameters of the experiment. The modified content is as follows: (Line 710-718)

Likewise, Ahmed et al. [110] also reported that under optimal conditions (600 MPa), high-pressure treatment alone markedly enhanced protein emulsification, with pH shifting playing a regulatory role in this process. Additionally, the synergistic application of pH shifting and high-pressure treatment substantially influenced the crystallization behavior and crystal morphology of protein, providing a molecular foundation for structural control and texture design in protein-based products [115]. Such modification has indicated broad applicability and promising potential across various plant proteins, including those derived from peas, rice dregs, soybeans, hemp seeds, and quinoa [67,116].

Comment 23:

23) Page 19, lines 695- 697: Review writing: Wang and Moraru reported that, during high-HPP treatment, lowering the pH or adding calcium promoted the formation of more porous and aggregated microstruc-696 tures in milk protein concentrate (MPC) [114].

Suggestion: Write the conditions of pH, HPH treatment and calcium concentration.

Response 23: Regarding the content you mentioned, after reviewing the original literature, we have supplemented it. The modified content is as follows: (Line 733-736)

Wang and Moraru [120] demonstrated optimal microstructural modification in milk protein concentrate (MPC) using pH 5.1 36 mg/g Ca combined with high-pressure homogenization (600 MPa, 5°C, 3 min), achieving 58% porosity increase and 3.2-fold aggregate size enhancement.

Comment 24:

24) Page 19: Improve Figure 5. It is not possible to read the legends of the figures (eixos x and y).

Response 24: Thank you for your comment. We have improved Figure 5 by increasing the resolution and enlarging the font size of the x- and y-axis labels to ensure that all legends are clearly readable. We have also adjusted the overall layout and visual clarity to enhance readability.

Figure 5. Effect of high pressure and pH treatment on the functionality and structure of various plant protein. Soybean protein isolate (A) [109]; hemp milk (B) [119]; rice dreg protein (C) [104]; hempseed protein isolate (D) [110].

Comment 25:

25) Page 20, lines 731- 746: Review writing: Earlier studies show that PEF treatment can markedly alter the secondary and tertiary structure of porcine myofibrillar proteins, along with their physicochemical properties [117]. By applying high-voltage pulses, PEF induced partial unfolding of protein molecules and exposes internal hydrolysis sites, thereby enhancing their enzymatic digestibility [118]. Compared with traditional thermal and high-pressure treatments, PEF can trigger conformational changes under milder conditions, facilitating protein digestion in the gastrointestinal tract [119]. Moreover, PEF improved protein hydrophilicity and dispersibility, while exposing additional reactive sites on amino acid residues, which facilitated interactions with other macromolecules such as polysaccharides and polyphenols [120]. In protein-polysaccharide complexes, PEF modulated intermolecular interactions, thereby affecting the emulsification, foaming, and gelling properties of the complexes [121]. Notably, the synergistic effects of PEF are significantly enhanced when applied in combination with pH-shifting treatment. While pH-shifting initiates conformational loosening by altering the protein’s charge distribution, PEF further promotes unfolding and the formation of smaller protein aggregates, thereby improving molecular flexibility and structural stability [122].

Suggestion: Write type of proteins, the conditions of pH-shifting and PEF treatment.

Response 25: Thank you very much for your valuable comment regarding the methodological clarity of the cited studies. We sincerely apologize for the oversight in not specifying the protein types and experimental conditions in the original manuscript. In the revised version, we have added the following details to ensure full transparency: (Line 775-795)

Earlier studies show that PEF treatment (2.1 kV/cm) for 5-30 min at 25 °C significantly altered the secondary and tertiary structures of porcine myofibrillar proteins, and emulsification properties (EAI: 7.10 m2/g, ESI: 179.67%) [123]. By applying high-voltage pulses at 1.00-1.25 kV/cm electric field strength, 20 μs pulse width, and 50 Hz frequency, PEF induced partial unfolding of bovine muscle proteins by disrupting Z-disk and I-band junctions, exposing internal hydrolysis sites (e.g., troponin T and myosin heavy chain), thereby improving in vitro protein digestibility by 18-31% (p < 0.05) after 180 min of simulated gastro-small intestinal digestion. [124]. Compared with traditional thermal and high-pressure treatments, PEF can trigger conformational changes under milder conditions, facilitating protein digestion in the gastrointestinal tract [125]. Moreover, PEF improved protein hydrophilicity and dispersibility, while exposing additional reactive sites on amino acid residues, which facilitated interactions with other macromolecules such as polysaccharides and polyphenols [126]. Notably, the synergistic effects of pulsed electric field (PEF) are significantly enhanced when applied in combination with pH shifting treatment. Under natural pH conditions (5.49-5.66) close to the isoelectric point of myofibrillar proteins, pH shifting initiates conformational loosening by altering protein charge distribution. When coupled with constant-current PEF (12-40 mA, 1000 Hz, needle-spring electrodes) or constant-voltage PEF (20 V/cm, 1000 Hz), the electric field further promotes protein unfolding and formation of smaller aggregates through suppression of disulfide bond formation and carbonyl compound accumulation (reducing oxidation by 18.3-28.7% compared to conventional thawing) [127].

Comment 26:

26) Page 20, lines 756-768: Review writing: Wang, et al. reported that the solubility of soy protein isolate increased from 26.06 to 70.34% following PEF treatment and pH shifting. Corresponding enhancements in emulsifying and foaming abilities further suggested that structural unfolding and optimized particle distribution support interfacial behavior [124]. Similarly, The combination of PEF and pH shifting, particularly at pH 4, effectively enhanced protein digestibility by improving the enzymatic hydrolysis of egg white protein while reducing protein aggregation, thereby increasing its functionality. [125].

Additionally, PEF-assisted pH-shifting treatment has been shown to increase the extraction yield and purity of plant-derived proteins while improving their microstructure and emulsion stability [126]. Yang et al. further observed that constant-current PEF thawing (CC-T) and pH shifting significantly improved the solubility, water-holding capacity, and thermal stability of frozen pork proteins, probably attributed to optimized protein structure and reduced oxidative damage.

Suggestion: Write the conditions of pH-shifting and PEF treatment.

Response 26: Thank you very much for your valuable comment regarding the methodological clarity of the cited studies. We sincerely apologize for the oversight in not specifying the experimental conditions in the original manuscript. In the revised version, we have added the following details to ensure full transparency: (Line 805-816)

For example, Wang et al. [129] reported that combined treatment with pH shifting (pH 11) and moderate PEF strength (10 kV/cm)synergistically increased the solubility of soy protein isolate from 26.06% to 70.34%. Corresponding enhancements in emulsifying activity and foaming ability were attributed to conformational unfolding, reduced particle size, and elevated surface hydrophobicity. These structural modifications optimized molecular flexibility and interfacial adsorption, thereby enhancing emulsifying and foaming behaviors. Similarly, The combination of PEF (1.4-1.8 kV/cm, 653-695 kJ/kg) and pH shifting, particularly at pH 4, effectively enhanced protein digestibility by improving the enzymatic hydrolysis of egg white protein while reducing protein aggregation, thereby increasing its functionality. [130]. Additionally, PEF pre-treatment (1.1 kV/cm, 1.53 kJ/kg) combined with pH shifting to5.0 during protein precipitation significantly increased the extraction yield of grass-clover juice by 25% and crude protein release by 31%. [131].

Comment 27:

27) Page 21, lines 780-781: Review writing: Zeng et al. successfully modified soy protein isolate using a PEF and pH-shifting to develop SPI-based nanoparticles for efficient lutein encapsulation [128].

Suggestion: Write the conditions of PEF treatment and pH-shifting.

Response 2: Thank you very much for your valuable comment regarding the methodological clarity of the cited study. We sincerely apologize for the oversight in not specifying the experimental conditions in the original manuscript. In the revised version, we have added the following details: (Line 828-830)

For example, Wang et al. [133] successfully modified soy protein isolate using a combination of pulsed electric field (10 kV/cm) and pH shifting to pH 11 to develop SPI-based nanoparticles for efficient lutein encapsulation.

Comment 28:

28) Page 21, line 794: Review writing: novel method for non-thermal sterilization in dairy applications [130].

Response 28: Thank you for your helpful comment. The content here is redundant and has been deleted.

Comment 29:

29) Page 22, lines 820-833: Review writing: Han et al. reported that the application of pH shifting and microwave heating to potato protein increased its solubility from 24.0 to 89.0%, surface hydrophobicity from 125 to 207, and reduced particle size from 249.95 μm to 90.37 nm, while simultaneously enhancing its dispersibility, emulsification, and thermal stability [132].

Similarly, Das et al. found that microwave-assisted extraction combined with pH modification optimized the yield and purity of soybean meal protein isolate, highlighting the synergistic effect of these techniques in enhancing extraction efficiency and quality control [133]. Jahan et al. further noticed that microwave (power 800, time 2 min) and pH shifting (pH 12) significantly improved the functional properties of mustard seed meal protein, thereby expanding its potential in plant protein processing [134]. Teixeira et al. showed that the synergistic effect of microwave and pH shifting can induce conformational rearrangement in proteins, regulate secondary structure composition, and significantly reduce protein aggregate size, thereby improving thermal stability and functional properties [135].

Suggestion: Write the pH and conditions of microwave treatment.

Response 29: Thank you for your helpful comment. We appreciate the suggestion to add the specific pH and microwave conditions. We have added these details in the revised manuscript: (Line 866-876)

For example, Han et al. [136] reported that the application of pH shifting (pH 12) and microwave heating (450W, 90 °C, 2/ 4/ 6 min) to potato protein increased its solubility from 24.0 to 89.0%, surface hydrophobicity from 125 to 207, and reduced particle size from 249.95 μm to 90.37 nm, while simultaneously enhancing its dispersibility, emulsification, and thermal stability. Similarly, Das et al. [137] found that microwave-assisted extraction (600 W, 32 s) combined with pH modification (pH 8) optimized the yield and purity of soybean meal protein isolate, highlighting the synergistic effect of these techniques in enhancing extraction efficiency and quality control. Jahan et al. [138] further noticed that microwave (power 800, time 2 min) and pH shifting (pH 12) significantly improved the functional properties of mustard seed meal protein, thereby expanding its potential in plant protein processing.

Comment 30:

30) Page 22, lines 851-857: Review writing: Zhang et al. [137] successfully optimized protein extraction from herring by-products using a combination of radial discharge high-shear homogenization and ultrasound-assisted pH shifting, while preserving antioxidant activity. This demonstrates the feasibility of synergistic enhancement between mechanical fields and pH modulation. Similarly, Igartúa et al. [138] combined pH shifting, ultrasound, and thermal treatment to modify rice proteins, resulting in increased solubility and surface hydrophobicity as well as the formation of loosely dispersed aggregates, which enhanced structural plasticity.

Suggestion: Write the pH and conditions of treatments.

Response 30: Thank you very much for your valuable comment. We sincerely appreciate your suggestion to specify the pH and treatment conditions mentioned in the examples. We have added these details to provide clearer information, including: (Line 891-912)

Building on this trend, an increasing number of studies have demonstrated the functional advantages of integrating physical, chemical, and enzymatic techniques to tailor protein properties for specific applications. Zhang et al. [139] optimized the extraction of proteins from herring by-products through the integration of radial discharge high-shear homogenization and ultrasound-assisted pH shifting treatment. The homogenization was conducted at 20,000 rpm for 3 minutes to disrupt cellular structures, followed by alkaline solubilization at pH 11.5 and subsequent isoelectric precipitation at pH 5.5. Ultrasound was applied at a power of 400 W for 10 minutes during the pH shifting process to enhance mass transfer and protein unfolding. This combined methodology not only improved protein recovery efficiency but also effectively retained the antioxidant activity of the extracts. These findings support the synergistic potential of mechanical and chemical treatments in modulating protein structure and functionality. Similarly, Igartúa et al. [140] investigated the structural and functional modification of rice protein isolates using a sequential treatment comprising pH shifting, high-intensity ultrasound, and thermal processing. The proteins were first solubilized under alkaline conditions at pH 11.0 to promote unfolding and dispersion. Subsequently, ultrasound treatment was applied using a probe-type sonicator (20 kHz) at full amplitude (100%) for 5 minutes to enhance mass transfer and facilitate molecular rearrangement. This was followed by thermal treatment at 85 °C for 30 minutes to stabilize the induced conformational changes. The combined treatment resulted in a marked increase in protein solubility and surface hydrophobicity, along with the formation of loosely associated aggregates. These structural alterations collectively contributed to improved plasticity and functional adaptability of the rice proteins.

Comment 31:

31) Page 23, lines 863-866: Review writing: Karabulut et al. applied a combination of thermal ultrasound, high-pressure homogenization, and pH shifting to hemp protein, which not only enhanced its solubility and emulsifying properties but also improved digestibility by loosening the protein structure [139].

Suggestion: Write the pH and conditions of treatments.

Response 31: We greatly appreciate your insightful feedback concerning the technical parameters in these referenced studies. We regret the omission of specific processing conditions in our initial submission. To address this, we have now incorporated comprehensive methodological details in the amended manuscript to provide complete experimental transparency: (Line 912-929)

Furthermore, Sun et al. [49] examined the gelation behavior of soy protein isolate (SPI) subjected to enzymatic modification with transglutaminase (TGase) under controlled alkaline and thermal conditions. The treatment was performed at pH 8.0 with TGase added at a concentration of 20 U/g protein, followed by incubation at 50 °C for 60 minutes to facilitate cross-linking. This process resulted in a significant enhancement of gel hardness and water-holding capacity, which was attributed to the formation of covalent bonds between glutamine and lysine residues, thereby strengthening the protein network structure. Similarly, Karabulut et al. [141] explored the functional modification of hemp protein isolates through a combination of thermal ultrasound, HPH, and pH shifting. Alkaline solubilization was conducted at pH 11.0 for 60 minutes to promote protein unfolding, followed by isoelectric precipitation at pH 4.5. Ultrasound treatment was applied using a 25 kHz probe at 400 W for 30 minutes under controlled heating at 60 °C. Subsequently, the protein dispersion underwent HPH at 100 MPa for three cycles to induce further particle size reduction and structural disruption. This multi-step treatment significantly improved the solubility and emulsifying properties of hemp protein and contributed to a more relaxed and accessible protein structure, thereby enhancing its potential functionality in food formulations.

Comment 32:

32) Page 23, lines 875-877: Review writing: Ding et 875 al. found that ultrasound-assisted pH-shifting significantly enhanced the interfacial adsorption capacity of the whey protein and carboxymethyl cellulose complex [141].

Suggestion: Write the pH and conditions of ultrasound treatment.

Response 32: We greatly appreciate your insightful feedback concerning the technical parameters in these referenced studies. We regret the omission of specific processing conditions in our initial submission. To address this, we have now incorporated comprehensive methodological details in the amended manuscript to provide complete experimental transparency: (Line 937-948)

在一项相关研究中,Ding 等人。 [143] 表明超声辅助 pH 值变化处理有效改善了乳清分离蛋白 (WPI)-羧甲基纤维素 (CMC) 复合物的界面吸附行为。处理包括将系统调节至 11.0 的碱性 pH 值以诱导蛋白质去折叠,然后酸化至 pH 5.0 以促进在 WPI 等电点附近形成复合物。以 450 W 的功率和 20 kHz 的频率应用超声 10 分钟,以增强分子相互作用和结构重排。这种方法显著提高了油-水界面的吸附能力,这可能是由于蛋白质和多糖组分之间的静电和氢键相互作用得到改善。这些发现强调了超声辅助 pH 值变化在乳化应用中调节蛋白质-多糖相互作用的潜力。

评论 33:

33) 第 25 页,第 958 行:评论写作:pH

回应 33:我们衷心感谢审稿人对我们稿件的细致和专业的评价 我们纠正了原始稿件中的书写错误。

Reviewer 2 Report

Comments and Suggestions for Authors

The manuscript presents a revision of how the pH-shifting and physical processing methods increase the functional properties of proteins.

This document needs cleaning up, but most of it reads well. For instance, it needs to be changed or improved:

  1. Figure 2 shows the effect of shifting the pH. This shows that the protein in its native conformation, before the pH shifting, thiol groups are in their free form and the shift to more basic pH promotes the formation of disulfide bonds. However, this is contradicted by what is stated in lines 266-267 t “Under extreme pH conditions, increased electrostatic repulsion among complex subunits leads to the disruption of hydrogen and disulfide bonds” and in lines 269-270 “alkaline pH-shifting induces the cleavage of disulfide bonds in gliadin complexes (with a 30% increase in free sulfhydryl groups)”.

  1. In general, the resolutions of the figures needs to be improved. It is difficult to read the text inside the figure. In addition, the meaning of abbreviations that appear in the figures is not indicated at the figure caption.

  1. A PPI abbreviation appears on line 402 and the meaning cannot be seen until line 420. Perhaps the meaning of the PPI can be added the first time that it appears in the text.

  1. On line 405 there is an abbreviation PPI whose meaning is missing in the text.

  1. Figure and table references are not included in the text. This makes it difficult for the reader to see the information they contain when reading the document.

  1. In table 1 the author used the abbreviation PPI for potato protein isolate and in the text of the document this abbreviation is used for pea protein isolate. Care should be taken not to confuse the reader. Be careful, because at the same table you are using the abbreviation PPI for pea protein isolate and perilla protein isolate as well. You should review all the abbreviations used throughout the document.

  1. On lines 630 and 631, there is an error. Perhaps it should be “Presently, the synergistic application of pH-shifting and high-pressure treatment, particularly HPH, has attracted significant attention in the field of protein modification” instead of “Presently, the synergistic application of pH-shifting and high-pressure treatment-particularly HPH-has attracted significant attention in the field of protein modification”

  1. On line 635 it should be HPH instead of (HPH).

  1. On line 691 it should be HPH instead of HHP. If not, you should report the meaning of this abbreviation.

  1. On line 695, 698 and 705 should be HPH instead of HPP. If not, you should report the meaning of this abbreviation.

  1. Sometimes pH-shifting is used and in other situations pH shifting is used. It is important to use the same format throughout the document.

  1. An abbreviation DHA appears on line 874 and its meaning is missing.

Author Response

Dear Editor and Reviewers,

Thanks very much for taking your time to review this manuscript. On behalf of all the authors, we appreciate the opportunity to resubmit our manuscript “Modification of food proteins by physical processing-assisted pH shifting techniques: A Comprehensive Review with revision. In general, we found the reviewer’s comments to be very constructive and insightful. The helpful criticism has enabled us to present a better and more informative manuscript.

We have presented the responses directly after each comment raised by the reviewer and the revised portions in the manuscript are marked in red. Our responses appear in italics below.

Responses to Reviewer #2:

General Response: We sincerely thank the reviewers for their meticulous, constructive, and insightful feedback on our manuscript. Your expertise has highlighted critical areas for enhancement, significantly strengthening the scientific rigor and clarity of our work. We have addressed all comments with utmost attention.

Comment 1:

  1. Figure 2 shows the effect of shifting the pH. This shows that the protein in its native conformation, before the pH shifting, thiol groups are in their free form and the shift to more basic pH promotes the formation of disulfide bonds. However, this is contradicted by what is stated in lines 266-267 t “Under extreme pH conditions, increased electrostatic repulsion among complex subunits leads to the disruption of hydrogen and disulfide bonds” and in lines 269-270 “alkaline pH-shifting induces the cleavage of disulfide bonds in gliadin complexes (with a 30% increase in free sulfhydryl groups)”.

Response 1: Thank you for your valuable suggestions. The contradiction points you raised are constructive. Based on your suggestions, I made modifications in Figure 2 and provided explanations below. Meanwhile, the descriptions of lines 266-267 and lines 269-270 have been supplemented. The disulfide bond is one of the important covalent bonds in protein structure and plays a key role in the stability, conformation and function of proteins. Most disulfide bonds in natural proteins are formed within folded tertiary or quaternary structures, formed by the thiol groups of cysteine residues in an oxidizing environment. They have a stable structure and are not easily affected by mild environmental changes.

When proteins are exposed to strongly acidic conditions (typically pH < 3.5), the conformation of the proteins undergoes significant changes. A large number of side chain groups protonate, and the originally stable hydrophobic and electrostatic interactions in the protein structure are disrupted or weakened, thereby exposing the disulfide bonds that were originally hidden within the proteins. Meanwhile, in an acidic environment, it may also promote the reduction or cleavage of disulfide bonds. Under alkaline shift conditions (typically pH > 10.5), the amino acid side chain groups of proteins deprotonate, the charged state of proteins changes, and the structure tends to be loose. Under strong alkaline conditions, sulfhydryl groups are prone to ionize to form sulfhydryl anions. The reactivity of these negatively charged sulfhydryl groups is enhanced, which may promote the disulfide bond exchange reaction, forming new disulfide bonds or causing the existing disulfide bonds to break. Furthermore, in a high pH environment, the protein structure becomes loose and the exposure of free thiol groups increases, which may lead to rearrangement of disulfide bonds, thereby forming new disulfide bonds or isomers, further affecting the function and stability of the protein.

In conclusion, acidic shift conditions may lead to the exposure, rupture or reduction of the originally stable disulfide bond structure of proteins, thereby significantly reducing the structural stability of proteins. Under alkaline shift conditions, disulfide bonds of proteins are prone to rearrangement and cleavage, which may generate novel conformations with structures and functions different from the original proteins.

Figure 2. Effect of pH shifting on primary (A), secondary (B), and tertiary (C) structure of protein.

lines 266-267 “Under extreme pH conditions, increased electrostatic repulsion among complex subunits leads to the disruption of hydrogen and disulfide bonds” replaced with Under extreme pH conditions, the increase in electrostatic repulsion between complex subunits leads to the destruction of hydrogen bonds. Meanwhile, the unstable structure also causes the breaking and recombination of disulfide bonds, promoting the dissociation and recombination of complexes. (Line 266-269)

Comment 2:

  1. In general, the resolutions of the figures needs to be improved. It is difficult to read the text inside the figure. In addition, the meaning of abbreviations that appear in the figures is not indicated at the figure caption.

Response 2: Thank you for your valuable feedback. We have carefully checked all figures to improve their resolution, ensuring that all textual elements are now clearly legible. Due to conversion to PDF, the image display may not be clear. We can ensure that the uploaded pictures will not cause confusion for readers. We sincerely appreciate your suggestion, which has helped improve the overall quality of the manuscript.

Comment 3:

  1. A PPI abbreviation appears on line 402 and the meaning cannot be seen until line 420. Perhaps the meaning of the PPI can be added the first time that it appears in the text.

Response 3: Thank you for your comment. We took your correct advice and added the meaning of PPI at an earlier stage. (line 404-407)

Wang et al. [56] reported the changes in the structure and gel properties of peanut protein isolate (PPI1) under the synergistic effect of temperature and pH shifting. The breaking force and water holding capacity of pH 10-treated PPI1 10-40 (at 40°C) gel were 2.2 times and 2.15 times higher than that of pH 7-adjusted sample at 25°C.

Comment 4:

  1. On line 405 there is an abbreviation PPI whose meaning is missing in the text.

Response 4: Thank you for your comment, we added the illustration of PPI. (Line 406-408)

Comment 5:

  1. Figure and table references are not included in the text. This makes it difficult for the reader to see the information they contain when reading the document.

Response 5: Thank you very much for pointing out this important issue.We have carefully reviewed the entire manuscript and ensured that all figures and tables are now properly referenced and cited within the main text at appropriate locations. These references have been inserted where the corresponding content is discussed, which we believe will significantly improve the readability and coherence of the manuscript. We are grateful for your thoughtful suggestion, which has contributed to the overall clarity and completeness of our work.

Comment 6:

  1. In table 1 the author used the abbreviation PPI for potato protein isolate and in the text of the document this abbreviation is used for pea protein isolate. Care should be taken not to confuse the reader. Be careful, because at the same table you are using the abbreviation PPI for pea protein isolate and perilla protein isolate as well. You should review all the abbreviations used throughout the document.

Response 6: Thank you for your comment. Your suggestion is very important. According to your suggestion, I checked the abbreviations of the entire text. For the different proteins with the abbreviation of PPI, such as, peanut protein isolate, pea protein isolate, perilla protein isolate and potato protein isolate we have respectively marked them as PPI 1, PPI 2, PPI 3, PPI4.

Comment 7:

  1. On lines 630 and 631, there is an error. Perhaps it should be “Presently, the synergistic application of pH-shifting and high-pressure treatment, particularly HPH, has attracted significant attention in the field of protein modification” instead of “Presently, the synergistic application of pH-shifting and high-pressure treatment-particularly HPH-has attracted significant attention in the field of protein modification”

Response 7: Thank you for your careful comment. I have replaced the original sentence. (Line 651-652)

Presently, the synergistic application of pH shifting and high-pressure treatment, particularly HPH, has attracted significant attention in the field of protein modification.

Comment 8:

  1. On line 635 it should be HPH instead of (HPH).

Response 8: Thank you for your nuanced comments. I have modified the details you mentioned and replaced (HPH) with HPH. (Line 659)

Comment 9:

  1. On line 691 it should be HPH instead of HHP. If not, you should report the meaning of this abbreviation.

Response 9: Thank you for your valuable comment. We have carefully revised the manuscript accordingly. (Line 736)

Comment 10:

  1. On line 695, 698 and 705 should be HPH instead of HPP. If not, you should report the meaning of this abbreviation

Response 10: We sincerely appreciate your insightful suggestion. In response, we have revised the corresponding section (Line 749)

Comment 11:

  1. Sometimes pH-shifting is used and in other situations pH shifting is used. It is important to use the same format throughout the document.

Response 11: Thank you very much for your valuable suggestion. To improve the consistency and academic accuracy of the manuscript, we have replaced all instances of “pH-shifting” with “pH shifting” throughout the text. We believe this revision enhances the clarity and uniformity of the terminology used in the manuscript. Comment 12:

  1. An abbreviation DHA appears on line 874 and its meaning is missing.

Response 7: Thank you for your valuable suggestions. We have made modifications and appropriate deletions (Line 940-942)

Reviewer 3 Report

Comments and Suggestions for Authors

1.Title is very complicated. Please make it simple so that reader can easily understand the aim of this review. 

2. The review manuscript should clearly indicate the aim in the abstract section. 

3. The grammar throughout the manuscript needs improvement. For example, the following sentence is not grammatically correct  "increasing demand for sustainable protein sources highlights the need for more efficient processing of traditional proteins, and the development of novel alternatives such as plant-and algae-derived proteins".

4. All verbs should consistently be written in the past tense.

5. The Introduction should provide a clearer rationale and better articulate the knowledge gap this study aims to address.

6. The authors should specify the pH shifting techniques method still now used. 

7. The manuscript should avoid the use of first-person pronouns such as “we,” “our,” and “us.”

8. The manuscript should accurately differentiate between present and past pH shifting techniques.

9. Figure 1. Keyword network diagram for protein functional properties research. please make proper explanation needed. 

10. Introduction is too much long and it is difficult to understand. Please make it concise and more information needed. Also please make it paragraph. 

11. Figure 2. what is the mechanism ? what is the massage from this figure ? why no description ?

12. How do you prepare this figure ? 

13. what is the citation ?

14. Do you have any evidence showing that you have permission from the author to use the photos in figures 3 and 4 in your manuscript? If you did not inform the author and publish the photos without permission, it is a copyright issue. you cannot only give the citation. 

15. To be honest, I did not find any novelty from your study. Please give more recent information and compare the novel idea. 

Author Response

Dear Editor and Reviewers,

Thanks very much for taking your time to review this manuscript. On behalf of all the authors, we appreciate the opportunity to resubmit our manuscript “Modification of food proteins by physical processing-assisted pH shifting techniques: A Comprehensive Review” with revision. In general, we found the reviewer’s comments to be very constructive and insightful. The helpful criticism has enabled us to present a better and more informative manuscript.

We have presented the responses directly after each comment raised by the reviewer and the revised portions in the manuscript are marked in red. Our responses appear in italics below.

Responses to Reviewer #3:

General Response: We extend our sincere gratitude for the constructive feedback, which has been instrumental in refining this review. We have comprehensively addressed all concerns through critical revisions focused on:

clarity and conciseness: streamlining the title and abstract to explicitly state the review’s aim, while condensing the introduction for enhanced focus and rationale.

Technical rigor: correcting grammatical inconsistencies, standardizing verb tenses, eliminating first-person pronouns, and clarifying distinctions between past/present pH-shifting techniques.

Visual and scholarly integrity: redesigning figures with detailed mechanistic annotations, securing proper permissions, citations for adapted content, and resolving copyright compliance.

Novelty emphasis: strengthening the knowledge gap analysis and integrating recent advancements to underscore the review’s unique contribution.

These revisions collectively elevate the manuscript’s coherence, accuracy, and academic value. We deeply appreciate the reviewers’ insights.

Comment 1:

  1. Title is very complicated. Please make it simple so that reader can easily understand the aim of this review.

Response 1: Thank you for your suggestion, we have modified the title:

Physical Processing-Assisted pH Shifting for Food Protein Modification: A Comprehensive Review

Comment 2:

  1. The review manuscript should clearly indicate the aim in the abstract section.

Response 2: Thank you for your comment. We have modified some parts of the abstract and explained the purpose that the article intends to express in the abstract part, including (line 23-31):

To address these limitations, this review explores the integration of pH shifting with physical processing techniques such as ultrasound, high-pressure processing, pulsed electric fields, and thermal treatments. Moreover, this review highlights the effects of these combined treatments on protein conformational transitions and the resulting improvements in functional properties such as solubility, emulsification, foaming capacity, and thermal stability. Importantly, they reduce reliance on extreme chemical conditions the need for extreme conditions, providing sustainability in industrial applications.

Comment 3:

  1. The grammar throughout the manuscript needs improvement. For example, the following sentence is not grammatically correct "increasing demand for sustainable protein sources highlights the need for more efficient processing of traditional proteins, and the development of novel alternatives such as plant-and algae-derived proteins".

Response 3: We appreciate your comment. We have made comprehensive revisions to the grammar of the entire text. Please refer to the revisions of the sentences you mentioned below. (Line 17-19) We have also made corresponding checks and modifications to the other sentences in the full text according to your suggestions.

The increasing demand for sustainable protein sources has intensified interest in improving the processing efficiency of traditional proteins and developing novel alternatives, particularly those derived from plants and algae.

Comment 4:

  1. All verbs should consistently be written in the past tense.

Response 4: Thank you very much for your insightful comment. In response to your suggestion, we have carefully revised the manuscript to ensure consistent use of the past tense throughout. All relevant verbs have been modified accordingly, and first-person constructions have been rephrased using passive voice where appropriate to maintain formal scientific tone and consistency. We appreciate your attention to this aspect of the writing, which has helped enhance the clarity and grammatical consistency of our manuscript.

Comment 5:

  1. The Introduction should provide a clearer rationale and better articulate the knowledge gap this study aims to address.

Response 5: We sincerely appreciate the insightful suggestion regarding the articulation of the research rationale and knowledge gap. we indicated the innovation points of this article compared with previous studies and the main issues it explains, including, (Line 146-158):

Therefore, this review aims to provide an updated and comprehensive overview of the synergistic application of pH shifting and physical modification techniques in the structural and functional transformation of food proteins, from both plant and animal sources. The review systematically elucidates the physicochemical mechanisms governing protein conformational changes under extreme pH conditions, and elucidates their effects on secondary, tertiary, and quaternary structures. The review further explores how the integration of pH shifting and other physical techniques such as ultrasonication, microwave treatment, and pulsed electric field, enhances key functional properties, including solubility, emulsification, gelling, foaming capacity, and bioactivity. Additionally, it discusses current and potential applications of these combined strategies in food processing. While critically addressing existing technological constraints, research gaps, and future directions of pH-physical combined strategies for protein valorization in the food and agricultural industries were discussed.

Comment 6:

  1. The authors should specify the pH shifting techniques method still now used.

Response 6: We sincerely thank the reviewer for highlighting this important methodological detail. In response, we have comprehensively revised the Introduction section. We have listed some previous reviews on the effects of pH shifts on proteins, including: (Line 112-128)

Moreover, Momen et al. [28] presented a comprehensive review on the application of alkaline-mediated treatments for both the extraction and functional modification of proteins from plant and animal sources, and reported that alkaline conditions irreversibly unfolded protein structure, thereby improving solubility, emulsification, gelation, and bioactive compound binding properties. Additionally, Lou et al. [29] discussed the molten globule state of proteins induced by pH shifting. Their work focused on the structural characteristics of this intermediate state and its relevance to food processing applications, emphasizing the potential of partially unfolded proteins in improving emulsification, flavor retention, and gelation. Recently, Sultan et al. [30] provided an extensive review on pH shifting technique for the extraction of plant-based proteins from various sources such as legumes, cereals, and oilseeds, and noticed that such method had the ability to produce protein isolates with improved solubility, emulsification, foaming, and gelation properties. Concurrently, they systematically summarized the effects of extraction and precipitation pH on both protein yield and functionality, highlighting optimal pH ranges for various protein types. Furthermore, this review briefly mentioned the effect of pH shift combined with ultrasound on the structure and functionality of proteins, but did not provide a detailed explanation.  

Comment 7:

  1. The manuscript should avoid the use of first-person pronouns such as “we,” “our,” and “us.”

Response 7: Thank you for your careful review and helpful comment.

In accordance with your suggestion, we have revised the manuscript to remove all first-person pronouns such as “we,” “our,” and “us.” The corresponding sentences have been rephrased in an objective and impersonal style to align with the formal tone expected in scientific writing. We appreciate your attention to this detail, which has helped improve the academic rigor of our manuscript.

Comment 8:

  1. The manuscript should accurately differentiate between present and past pH shifting techniques.

Response 8: We sincerely thank the reviewer for this insightful critique regarding the methodological delineation of pH shifting techniques. We acknowledge that the original manuscript did not sufficiently clarify the evolution of these methods. To address this, we have discussed the past pH shift technology and the current pH technology in three sections. The first section mainly summarizes the research on the impact of pH shift technology on proteins in recent years. The second section clarifies some limitations of their research. The third section introduces the supplementary content and innovation points of this review, including: (Line 112-158):

Moreover, Momen et al. [28] presented a comprehensive review on the application of alkaline-mediated treatments for both the extraction and functional modification of proteins from plant and animal sources, and reported that alkaline conditions irreversibly unfolded protein structure, thereby improving solubility, emulsification, gelation, and bioactive compound binding properties. Additionally, Lou et al. [29] discussed the molten globule state of proteins induced by pH shifting. Their work focused on the structural characteristics of this intermediate state and its relevance to food processing applications, emphasizing the potential of partially unfolded proteins in improving emulsification, flavor retention, and gelation. Recently, Sultan et al. [30] provided an extensive review on pH shifting technique for the extraction of plant-based proteins from various sources such as legumes, cereals, and oilseeds, and noticed that such method had the ability to produce protein isolates with improved solubility, emulsification, foaming, and gelation properties. Concurrently, they systematically summarized the effects of extraction and precipitation pH on both protein yield and functionality, highlighting optimal pH ranges for various protein types. Furthermore, this review briefly mentioned the effect of pH shift combined with ultrasound on the structure and functionality of proteins, but did not provide a detailed explanation.

Although these studies have laid a foundation for understanding protein behavior under alkaline or pH-induced conditions, several limitations persist. including negligible functional improvement and extreme operational conditions. Notably, previous reports have largely overlooked the synergistic effects of combined pH shifting and physical processing techniques, such as high-intensity ultrasound (HIU), pulsed electric field (PEF), and high-pressure treatment. These emerging hybrid strategies may enable more precise control over protein unfolding, aggregation, or reconfiguration, thereby augmenting their functional characteristics. While, pH shifting significantly enhances functional properties-such as solubility and emulsifying activity via structural changes (e.g., β-sheet dissociation, hydrophobic group exposure), its potential limitations should not be overlooked. Extreme pH conditions may induce chemical modifications of amino acid residues (e.g., serine, lysine), stimulating reduced nutritional quality and the formation of toxic compounds like lysinoalanine (LAL). While alkaline extraction can temporarily enhance the solubility of rice protein, overexposure to high pH levels (pH > 12) may result in irreversible deamidation and the loss of essential amino acids [5]. Therefore, precise control of the pH range and the development of multiscale synergistic modification techniques have become critical research directions.

Therefore, this review aims to provide an updated and comprehensive overview of the synergistic application of pH shifting and physical modification techniques in the structural and functional transformation of food proteins, from both plant and animal sources. The review systematically elucidates the physicochemical mechanisms governing protein conformational changes under extreme pH conditions, and elucidates their effects on secondary, tertiary, and quaternary structures. The review further explores how the integration of pH shifting and other physical techniques such as ultrasonication, microwave treatment, and pulsed electric field, enhances key functional properties, including solubility, emulsification, gelling, foaming capacity, and bioactivity. Additionally, it discusses current and potential applications of these combined strategies in food processing. While critically addressing existing technological constraints, research gaps, and future directions of pH-physical combined strategies for protein valorization in the food and agricultural industries were discussed.

Comment 9:

  1. Figure 1. Keyword network diagram for protein functional properties research. please make proper explanation needed.

Response 9: We sincerely appreciate the valuable suggestion regarding the interpretation of Figure 1. According to your request, we have supplemented the relevant content: (Line 72-85)

In recent years, significant research efforts have been directed towards understanding the intricate relationship between pH shifts and the functional properties of proteins. This network visualization encapsulates multifaceted interactions and dependencies among various factors influencing protein functionality. At the core of this network lies the concept of “functional property,” which is intricately linked to several key attributes such as solubility, surface hydrophobicity, particle size, and emulsion stability. The functional role of proteins in food systems and industrial applications is determined by these properties. The network highlights the pivotal role of “solubility” as a central node, underscoring its importance in protein functionality. Solubility is influenced by multiple factors including pH, ionic strength, temperature, and post-translational modifications. The impact of pH on protein solubility is particularly noteworthy, as it directly affects the protein’s net charge, conformation, and subsequent interactions with other molecules. Overall, this network serves as a visual representation of the complex interplay between pH shifts and protein functional properties.

Comment 10:

  1. Introduction is too much long and it is difficult to understand. Please make it concise and more information needed. Also please make it paragraph.

Response 10: We are deeply grateful for the reviewer's constructive feedback on improving the clarity and conciseness of the Introduction. We have conducted a comprehensive review of the introduction part and deleted the repetitive content to make it as concise as possible. The modified parts were marked in red in the original manuscript.

Comment 11:

  1. Figure 2. what is the mechanism ? what is the massage from this figure ? why no description ?

Response 11: Thank you very much for your valuable comment regarding Figure 2. We sincerely apologize for the oversight in not providing a clear description of the mechanism and key message conveyed by this figure in the original manuscript.In the revised version, we have now added a detailed explanation of the underlying mechanism illustrated in Figure 2including: (Line 194-203)

Under normal conditions, the primary structure of proteins remains stable after a pH shift, and the order of amino acids remains basically unchanged. However, under extreme pH shift conditions, the secondary structure usually shows that both the α-helix and β-fold undergo structural deconvolution and destruction of hydrogen bonding. There is also structural loosening and rearrangement, as well as an increase in random coils. Significant changes occur to the tertiary structure of proteins under pH shift conditions. The tertiary structure of proteins undergoes significant changes under pH shift treatment, mainly manifested by the destruction of salt bridges within the protein molecule and the exposure of hydrophobic groups and sulfhydryl (-SH) groups, which can form disulfide bonds (-S-S-) [33].

Comment 12:

  1. How do you prepare this figure ?

Response 12: Thank you very much for your thoughtful question. This figure is a simplified mechanistic illustration that I created based on a review of relevant literature concerning the effects of pH shifts on proteins, particularly from the perspective of how pH changes impact protein structure. It primarily highlights the influence of pH-shift technology on the primary, secondary, and tertiary structures of proteins. According to your suggestion, I revised the figure again.

Figure 2. Effect of pH shifting on primary (A), secondary (B), and tertiary (C) structure of protein.

Comment 13:

  1. what is the citation ?

Response 13: Thank you very much for your careful reading and helpful comment. We apologize for the omission of the relevant citation in the original manuscript. We added references respectively in line (1294) and line (203).

Comment 14:

  1. Do you have any evidence showing that you have permission from the author to use the photos in figures 3 and 4 in your manuscript? If you did not inform the author and publish the photos without permission, it is a copyright issue. you cannot only give the citation.

Response 14: We sincerely thank the reviewer for emphasizing the importance of copyright compliance. We confirm that written permissions have been secured for all copyrighted images in Figures 3, 4, and 5. Some of the permitted image data are as follows. If you need specific files, I will send them to the editor immediately.

Comment 15:

  1. To be honest, I did not find any novelty from your study. Please give more recent information and compare the novel idea.

Response 15: Thank you very much for your candid and constructive feedback. We sincerely appreciate your comments regarding the novelty of our study. We understand your concern, and in response, we have carefully revised the manuscript to more clearly emphasize the novel aspects of our work.

By reviewing recent literature on pH-shift treatment of proteins, it was observed that relatively limited attention has been given to the combination of physical processing methods with pH-shift techniques. However, numerous recent studies have demonstrated that integrating physical processing methods with pH-shift treatment yields superior effects compared to using pH-shift treatment alone. Therefore, from the perspective of their combined influence on proteins, this paper provides a comprehensive summary of how such synergistic approaches affect the structural and functional properties of proteins. The novelty and supplementary contributions of this study, in comparison with previous research, are clarified in (line 146 to 158) of the manuscript.

Therefore, this review aims to provide an updated and comprehensive overview of the synergistic application of pH shifting and physical modification techniques in the structural and functional transformation of food proteins, from both plant and animal sources. The review systematically elucidates the physicochemical mechanisms governing protein conformational changes under extreme pH conditions, and elucidates their effects on secondary, tertiary, and quaternary structures. The review further explores how the integration of pH shifting and other physical techniques such as ultrasonication, microwave treatment, and pulsed electric field, enhances key functional properties, including solubility, emulsification, gelling, foaming capacity, and bioactivity. Additionally, it discusses current and potential applications of these combined strategies in food processing. While critically addressing existing technological constraints, research gaps, and future directions of pH-physical combined strategies for protein valorization in the food and agricultural industries were discussed.

Round 2

Reviewer 2 Report

Comments and Suggestions for Authors

The article can be published, the document has been improved after comments. The only thing missing is the references to figures 3 and 4 in the text, at least I haven't seen them. Otherwise, it looks good.

Author Response

We have carefully reviewed the manuscript content in line with your recommendations and promptly made the necessary revisions. Specifically, we have enhanced the explanations for Figures 2 and 3, ensuring clarity and coherence.

Reviewer 3 Report

Comments and Suggestions for Authors

Now many things already improved, still many thing need to mentioned. 

  1. What is the aim of the fig 1 ? please explain with fig title. 
  2. Fig 3 & 4, I again said that do you have any permission from author to use those fig ?
  3. Actually the review is too big to read. You need to rewrite and concise. 
  4. You used 143 references, do you really study all of those paper ? 
  5. Fig 5, it would be interesting if you enlarge the figure. 

Author Response

Dear Editor and Reviewers,

Thanks very much for taking your time to review this manuscript. On behalf of all the authors, we appreciate the opportunity to resubmit our manuscript “Physical Processing-Assisted pH Shifting for Food Protein Modification: A Comprehensive Review

” with revision. In general, we found the reviewer’s comments to be very constructive and insightful. The helpful criticism has enabled us to present a better and more informative manuscript.

We changed all the red marks modified in the first version to black. We have presented the responses directly after each comment raised by the reviewer and the revised portions in the manuscript are marked in red. Our responses appear in italics below.

Responses to Reviewer :

General Response: Thank you very much for your valuable comments and suggestions.  We have carefully revised the manuscript according to your feedback.  Major issues regarding figure clarification, content conciseness, reference use, and figure permissions have been addressed.  We believe these revisions have significantly improved the quality and clarity of the manuscript.

Comment 1:

  1. What is the aim of the fig 1 ? please explain with fig title

Response 1: We sincerely thank the reviewer for this helpful suggestion. Figure 1 mainly reflects the connections among the relevant literatures on keywords in recent years. According to your suggestion, we have streamlined the introduction part, deleted Figure 1 and the introduction about Figure 1, and also removed some parts with redundant descriptions. This indeed makes the introduction part simpler and clearer.

Comment 2:

  1. Fig 3 & 4, I again said that do you have any permission from author to use those fig ?

Response 2: Thank you very much for your rigorous and professional comment. We are very sorry that we failed to clearly inform you that we have obtained the copyright of the relevant pictures. We have obtained the Copyrights of fig2, 3 and 4 through application. To clearly display the license information, I list them one by one as follows

Fig 2(A)

Fig 2(B)

Fig 2(C)

Fig 3(A)

Fig 3(B)

Fig 3(C)

Fig 3(D)

Fig 4(A)

Fig 4(B)

Fig 4(C)

Fig 4(D )

If you need more specific relevant proof, please do let me know.

Comment 3:

  1. Actually the review is too big to read. You need to rewrite and concise..

Response 3: We appreciate your insightful comment. We have carefully revised the manuscript to improve its conciseness and readability. Redundant content has been reduced, and overly detailed sections have been streamlined to enhance clarity while retaining essential information. First of all, we deleted and streamlined the introduction part to make it more straightforward and clear. Secondly, we removed the content of "Effect of pH shifts on protein complexes" in the second part, making the overall content simpler and easier to understand; In addition, we have streamlined other redundant parts of the full text and the list of references.

Comment 4:

  1. You used 143 references, do you really study all of those paper ?

Response 4: Thank you for raising this important point. Indeed, we carefully reviewed a substantial number of references to ensure comprehensive coverage of the topic. However, considering your valuable feedback regarding manuscript length , we have streamlined the manuscript by removing some content and consequently reducing the number of references accordingly. We finally determined the references to 137.

Comment 5:

  1. Fig 5, it would be interesting if you enlarge the figure.

Response 5:Thank you for the helpful suggestion. Considering that the layout of the pictures might not be reasonable enough, we redesigned the pictures. We have reformatted and enlarged Figure as requested to improve its visual clarity and ensure that all relevant details are more easily visible to the reader.

Figure.4 Effect of high pressure and pH treatment on the functionality and structure of various plant protein. Soybean protein isolate (A) [109]; hemp milk (B) [119]; rice dreg protein (C) [104]; hempseed protein isolate (D) [110].
